# The Central Role of Iron in Human Nutrition: From Folk to Contemporary Medicine

**DOI:** 10.3390/nu12061761

**Published:** 2020-06-12

**Authors:** Matteo Briguglio, Silvana Hrelia, Marco Malaguti, Giovanni Lombardi, Patrizia Riso, Marisa Porrini, Paolo Perazzo, Giuseppe Banfi

**Affiliations:** 1IRCCS Orthopedic Institute Galeazzi, Scientific Direction, 20161 Milan, Italy; banfi.giuseppe@fondazionesanraffaele.it; 2Department for Life Quality Studies, University of Bologna, 47921 Rimini, Italy; silvana.hrelia@unibo.it (S.H.); marco.malaguti@unibo.it (M.M.); 3IRCCS Orthopedic Institute Galeazzi, Laboratory of Experimental Biochemistry and Molecular Biology, 20161 Milan, Italy; giovanni.lombardi@grupposandonato.it; 4Department of Athletics, Strength and Conditioning, Poznań University of Physical Education, 61-871 Poznań, Poland; 5Department of Food, Environmental and Nutritional Sciences (DeFENS), Division of Human Nutrition, University of Milan, 20133 Milan, Italy; patrizia.riso@unimi.it (P.R.); marisa.porrini@unimi.it (M.P.); 6IRCCS Orthopedic Institute Galeazzi, Postoperative Intensive Care Unit & Anesthesia, 20161 Milan, Italy; paoloperazzo1@virgilio.it; 7Faculty of Medicine and Surgery, Vita-Salute San Raffaele University, 20132 Milan, Italy

**Keywords:** iron, anemia, vitamin, dietary supplements, nutraceutical, functional food, integrative medicine, preoperative care, transfusion-alternative strategy, elective surgical procedures

## Abstract

Iron is a fundamental element in human history, from the dawn of civilization to contemporary days. The ancients used the metal to shape tools, to forge weapons, and even as a dietary supplement. This last indication has been handed down until today, when martial therapy is considered fundamental to correct deficiency states of anemia. The improvement of the martial status is mainly targeted with dietary supplements that often couple diverse co-factors, but other methods are available, such as parenteral preparations, dietary interventions, or real-world approaches. The oral absorption of this metal occurs in the duodenum and is highly dependent upon its oxidation state, with many absorption influencers possibly interfering with the intestinal uptake. Bone marrow and spleen represent the initial and ultimate step of iron metabolism, respectively, and the most part of body iron circulates bound to specific proteins and mainly serves to synthesize hemoglobin for new red blood cells. Whatever the martial status is, today’s knowledge about iron biochemistry allows us to embrace exceedingly personalized interventions, which however owe their success to the mythical and historical events that always accompanied this metal.

## 1. Etymology and Ages of Iron Myths

Mars, the god of war for the ancient Romans, had to be worshiped prior to battle by believing soldiers. Known as Ares by Greeks, he was said to love bloody battles. Since the Hittites of Near East and other metalsmiths in the Mediterranean region began to learn how to heat iron with charcoal powder, harder and more durable weapons than previous bronze or iron-only counterparts had undoubtedly made battles fiercer. These discoveries occurred around 1200 B.C. after the collapse of the Bronze Age kingdoms, which gave rise to the so-called Iron Age [1,2].

Iron was easy to find and extract, as it is one of the most abundant elements on our planet, being constitutive of both the inner and outer cores of Earth and its surface crust since the beginning of the world. The chemical symbol Fe comes from the Latin name of iron, *ferrum*, and it is placed in the periodic table among transition metals with coordinates group 8, block d, and period 4. Still existing as full energy form in the core of supernova remnants, common oxidation states of iron are −2, 0, 2 (Fe^2+^, ferrous status), 3, and 6, with the ferric status (Fe^3+^) being the most stable on Earth’s aerobic atmosphere and neutral pH environments [3]. Nevertheless, this prevalent form remains unavailable for organisms, which evolved to acquire diverse mechanisms for incorporating the metal, such as the reduction of Fe^3+^ to Fe^2+^. The inorganic oxide presents as a solid matter, shiny and silvery, hard and dense, good conductor of heat, often forming colored compounds. Probably, the dim reddish-orange appearance inspired the Romans to name the Red Planet like their god of war. Egyptians referred to it as “Her Desher”, which means “the red one” [4]. Indeed, the iron-containing dust or rust of the planet makes it appear mostly red like blood, thus mirroring the unpleasant violence of battles to ancient peoples. From the medieval alchemist’s symbolism, iron had always been represented as an oblique arrow originating from a circle (♂), which had been also used by astrologist to represent Mars, with his shield and spear, and nowadays by scholars to refer to the male gender as a Carl Linnaeus’ heritage [5].

## 2. The Martial Status in Humans

Although the evolution of biological beings diverged millennials ago, iron had been incorporated by plants and animals in an exceptionally similar manner in order to exploit its role in respiration. Plants acquired two key mechanisms: strategy I (H^+^ extrusion to promote Fe^3+^ reduction) and strategy II (release of specific Fe^3+^-chelating phytosiderophores and subsequent high-affinity uptake) [6]. In *Homo sapiens*, the complex regulation of deposits allocation (↑ release if ↑ requirements) and erythrocytes (RBCs) cycle assures the metal to be conserved in complex forms bound to proteins, with the intestinal passage (↑ absorption if ↓ reserves) being the main regulatory point of body iron balance. These mechanisms assure a total body iron of about 2.3 g in women and 3.8 g in men, with almost 60–70% being incorporated in the main circulating protein hemoglobin (Hb), 20% in iron deposits of ferritin, and about 15% in other proteins, mostly myoglobin in muscle tissue together with heme and non-heme enzymes and the iron transport protein transferrin (Tf) [7]. The blood content of the metal refers to a subject’s martial status (from Italian *profilo marziale*: *profilo* “profile” and *marziale* “martial”). This allegory still reminds of physical strength and stamina of the Roman god of war, and it is reasonable given the role of these iron-containing proteins in supplying tissues with oxygen. Despite the evolved biological strategies to incorporate iron from environments, both humans and plants commonly suffer from iron deficiency syndromes [8], which refer to the most common form of “anemia” (from Greek *ἀναιμία*: *ἀν*- “without” and -*αἷμα* “blood”) that is known to affect a third of the world’s population. Of note, children and pregnant women of the poorest regions of the world represent 55% of all anemia cases [9], which derive from the coexistence of physiological increased needs in conditions of both low bioavailability (e.g., cereal-based diet) and other causal factors, such as poverty, hookworm infections, and schistosomiasis [10]. Anemia was already acknowledged in the past when the lack of energy that a soldier could feel before the war could make a difference between victory and defeat. The hemorrhagic anemia often happened after battles and now easily occurs after surgeries. Nowadays, it is known that anemia is ascribed to different conditions, possibly mirroring a dysfunction of hematopoietic organs, hereditary diseases (e.g., sickle cell disease), or secondary conditions (e.g., vitamin B12 deficiency in pernicious anemia).

## 3. Folk Medicine

“In about 400 B.C. Hippocrates, the father of modern medicine, is supposed to have founded his first hospital beside a stream so that he could have watercress beds close by to boost his patients’ recovery” [11]. Indeed, watercress (*Nasturtium officinale* B.), which is part of the cruciferous vegetables, has an iron content (4 mg/100 g) comparable to that of spinach and a concentration of ascorbic acid (145 mg/100 g) higher than that of lemon (129 mg/100 g), greatly favoring iron absorption (see Section 4.2). These properties could be associated to the legendary consumption of watercress to acquire stamina before battles, on behalf of the Greek general Xenophon [11]. Iron had been used in folk medicine for millennia, and often for other purposes than mere bloody ones. Around 3500 B.C. ancient Egyptians used iron powder for baldness [12], which is now more elegantly named non-scarring alopecia. Reflecting on the tradition of physical strength associated with Ares, Greeks used a mixture of wine and iron to treat male impotency [12], thus possibly taking advantage of alcohol effects in augmenting self-confidence and dodging psychogenic inability. Among the consistent grandma’s remedies, the consumption of a tablespoon of blackstrap molasses (the viscous by-product from sugar cane or beets) is supposed “to pump more iron into your body” [13]. This could reflect the remarkable content of iron (5 mg/100 g) and the presence of some absorption enhancers which may help in correcting iron deficiency [14]. Despite nowadays-ethical issues, the tradition of eating horse liver (average of iron: 10–20 mg/100 g) marinated few hours in lemon juice (average of ascorbic acid: 99 mg/100 g) is still common among the elderly of southern Italy. They believe that this preparation could sustain the recovery from illness and tiredness, which is indeed often associated with low blood iron.

## 4. Current Knowledge on Iron Homeostasis

### 4.1. Overview of Iron Metabolism

#### 4.1.1. Gastric Processing

Upon ingestion, food is mixed with gastric juice to obtain proper solubilization of different micronutrients. In particular, iron needs to be reduced and prevented to form insoluble complexes upon chelation with low molecular weight substances. Several positive or negative reactions can occur at the level of the stomach (see Section 4.2) that can prevent the appropriate solubilization of the inorganic iron, thereby influencing its bioavailability upon entrance of the proximal small intestine [15]. Iron absorption mostly occurs at the level of the duodenum, close to the pyloric proximity in order to exploit the residual acidity before pH buffering.

#### 4.1.2. Intestinal Passage

The ferrireductase duodenal cytochrome b (DCYTB) helps to keep iron reduced at the absorption site, thus allowing the internalization through the divalent metal transporter (DMT1) [16], which is part of the family of proton-coupled metal ion transporters (SLC11A2). Concerning heme-iron, it is not yet clear how it can be internalized into the enterocyte [17]. The low-affinity heme carrier protein (HCP) has been proposed to have a role, with the metal being subsequently freed from the porphyrin ring by a heme oxygenase [18]. On the apical border, also dietary ferritin may be absorbed through endocytosis and then subjected to lysosomal digestion [19]. If there is positive iron homeostasis, reduced iron can be complexed with apoferritin to form ferritin deposits. If iron is required, the basolateral transporter ferroportin (SLC40A1) exports ferrous iron that is subsequently incorporated into apotransferrin by either the membrane-bound hephestin (copper-dependent ferroxidase, so named from “Hephaestus”, the Greek god of metalworking) or the circulating ceruloplasmin (ferroxidase produced by the liver) [20].

#### 4.1.3. Systemic Delivery

Tf binds all iron circulating in plasma and represents the most dynamic compartment, with a turnover rate of about ten times a day that meets the erythropoiesis requirements [21]. The complex Tf-iron interacts with a ubiquitously located receptor and is then internalized through receptor-mediated endocytosis [22]. The subsequent acidification of the vesicle lumen by proton pumps allows the offloading of iron-bound Tf and the entry in two different pathways: a recycling pathway, which implies recycling of Tf back to the plasma membrane for iron reloading, and the endosomal degradation pathway, which ends with the release of iron from the endosome thanks to SLC11A2 [23]. Iron can then be sequestered in the iron storage protein, ferritin, when it is not required for incorporation into functional iron proteins (heme, non-heme, Fe-S proteins) [24]. The iron regulatory proteins (IRP1 and IRP2) regulate cellular iron homeostasis by regulating iron uptake, utilization, and storage, which may be inferred from the concentration of circulating ferritin as it is normally secreted by cells in quantities proportional to intracellular deposits. The less prevalent hemosiderin is another iron storage complex that less easily releases the metal upon increased requirements [24].

#### 4.1.4. Physiological Roles

Hematopoietic tissues incorporate most of the blood iron, whereas other tissues, such as myocytes, internalize smaller quantities. In fact, erythrocyte precursors in the bone marrow of vertebrae, sternum, and ribs, highly express Tf receptors together with that of the kidney erythropoietin (EPO), thereby boosting the differentiation of cells that are part of the erythropoietic lineage during hypoxic conditions [17]. Of note, effective erythropoiesis requires folate and cobalamin to sustain the pyrimidine synthesis, with the terminal enzyme of the heme biosynthetic pathway (i.e., the ferrochelatase) having a key role in catalyzing the insertion of Fe^2+^ into the proto-porphyrin ring structure to form the heme molecule [24]. In the reticuloendothelial system of the human spleen, resident macrophages of the red pulp are in charge of senescent RBCs clearance [25], being capable of metabolizing hemoglobin through proteolysis, heme through heme oxygenase activity, and ferritin through lysosomal degradation. Unless it is not required, the metal exits the macrophages thanks to the SLC40A1, is oxidized by ceruloplasmin and bound to Tf, thus subsequently replenishing most of the Tf iron pool [26].

#### 4.1.5. Homeostatic Regulation

Other than the abovementioned IRP system, which mainly controls cellular iron uptake and deposits, there is also a general regulatory system for iron homeostasis. Primarily produced by hepatocytes, hepcidin is the master regulator that coordinates dietary absorption, storage, and tissue distribution [27]. Increased hepcidin reduces the number of exposed SLC11 and SLC40, thus blocking the intestinal passage. Consequently, it affects the release of iron from macrophages and hepatocytes, the latter having a great capability for iron deposition in the ferritin form [28]. Reduced iron entry into the bloodstream results in low Tf saturation and lesser iron to be delivered to tissues that expose Tf receptors. Dysregulation of these mechanisms results in iron disorders. Anemia from chronic disease is known to be associated with overexpression of hepcidin, macrophage iron loading, low blood iron, and reduced erythropoiesis [29]. Conversely, negligible hepcidin expression causes higher iron entry into the bloodstream, high Tf saturation, and excess iron accumulation in vital organs (e.g., hemochromatosis) [30].

### 4.2. Absorption Influencers

Iron easily changes its state of oxidation to form coordination complexes with other atoms capable of donating electrons, and some components named absorption influencers can frustrate or potentiate the intestinal passage. These influencers can be disruptors (i.e., negative effectors) or enhancers (i.e., positive effectors). For instance, well-known disruptors are specific gastrointestinal conditions, such as peptic ulcer diseases or even *Helicobacter pylori* gastritis [31]. A slowly bleeding from an ulcer that goes unnoticed may cause hemorrhagic anemia whereas chronic gastritis at the level of the body can cause an acid output reduction. Similarly, *H. pylori* infection leads to a reduction of the levels of L-ascorbic acid in the digestive fluid juice and some strains of this infective agent are even able to compete with the host for binding iron [32]. In addition, some medications, such as antacids, are known to substantially reduce iron absorption because of the lumen acidity neutralization, which is known to prevent the reduction of inorganic oxides. Other negative effectors are mainly of dietetic origin and form insoluble salts in the stomach, such as tannins, oxalic and phytic acids, polyphenols, or compete for/inhibit absorption, such as manganese, zinc, lead [33], and calcium [34]. Conversely, positive effectors are fructose, copper, vitamin A, and β-carotene, with the main absorption enhancer among all being the L-ascorbic acid. This water-soluble vitamin (daily needs for adults: 95–110 mg), historically indicated for the prevention and treatment of scurvy, has a reducing potential able prevent the oxidation of neighbouring molecules. It is known to exert positive pharmaceutical actions in the lumen of the stomach and small intestine by reducing non-heme Fe^3+^ to Fe^2+^ and acting as weak chelator, similarly to citric and lactic acid, to help solubilizing the metal. In cells, L-ascorbic acid can promote the release of iron from deposits.

### 4.3. Diagnostics of Iron Deficiency

#### 4.3.1. Understanting the Iron Deficiency

Anemia from iron deficiency is the most common anemia type [35] and may derive from inadequate intake (e.g., poor diet quality), malabsorption (e.g., gastritis, celiac disease, gastritis, gastrointestinal resection, iron refractory iron deficiency anemia), increased physiological requirement (e.g., growth, menses, pregnancy), or pathological blood loss (e.g., internal bleedings, menorrhagia, intravascular hemolysis). The nutritional iron deficiency is the most common cause of iron deficiencies and is mainly triggered by increased needs not fully guaranteed by dietary intakes [36]. This condition is eventually associated with a detectable change in different laboratory tests [37,38]. In 2007, a joint assessment of the WHO and the Centers for Disease Control and Prevention (CDC) indicated ferritin as a primary measure of the martial status at the population level and the soluble Tf receptor (sTfR) as a second promising parameter that warranted continued evaluation [39]. These two biomarkers are useful to categorize the anemia type as both mirror the intracellular iron homeostasis. As abovementioned, small quantities of ferritin are present in the serum reflecting the amounts deposited in cells. Similarly, small amounts of sTfR derive from the extracellular cleavage of the Tf receptor, and increased serum levels mirror negative iron homeostasis [40,41]. Nevertheless, ferritin is also an acute-phase protein involved in the inflammatory response against pathogens therefore being of limited use during infective and inflammatory conditions, but also in case of liver disease, tumor, hyperthyroidism, and heavy alcohol intake [42]. If not properly assessed, the prevalence of iron deficiency may be underestimated [43], as ferritin increases during inflammatory conditions irrespective of the martial status [44]. Consequently, it has been suggested to rise the cut-off value from 12 to 30 μg/L since an adjustment of ferritin values according to the individual’s inflammatory status has found no consensus yet. The sTfR is less influenced by inflammation, but other acute-phase mechanisms, such as hypoxia or iron-limited erythropoiesis, are known to possibly affect its circulating levels [45]. Regardless of the etiology, frank anemic conditions represent risk factors for bad conditions, especially in fragile individuals undergoing orthopedic surgery [46,47], and specific diagnostic algorithms are available to categorize the type to properly tailoring the intervention.

#### 4.3.2. The Martial Status Biomarkers during Iron Deficiency

The depletion of storages, iron-deficient erythropoiesis, and iron-deficient anemia are the increasingly severe consequences that arise upon iron deficiency, with the affection of erythroid cell development and feature being acknowledged by impaired RBCs homeostasis but even intracellular iron-containing proteins [48]. Although the measurement of blood parameters relies on well-established and widely used analytical methods, many concerns persist regarding the pre-analytical phase management and assay comparability/standardization.

*Iron storage depletion*. During the first phase of iron depletion, the deposits in the bone marrow, liver, and spleen are becoming exhausted (no stainable bone marrow iron), but no consequences on erythropoiesis are detectable yet. This early depletion is characterized by low ferritin (<35 μg/L), but normal Hb and other martial status indices [36]. The bone marrow is a major site for iron storage, but all the local metal is used for erythropoiesis, easily impairing RBC generation upon iron depletion at this site. The absence of stainable iron in the bone marrow is the gold standard for iron deficiency diagnosis, but it is used only in certain circumstances due to the invasive nature of the procedure [49]. It is based on the Prussian blue staining of aspirates to detect both hemosiderin in macrophages and iron granules in sideroblasts. The analysis requires an experienced observer and careful attention to detail [50]. The serum fraction of ferritin represents a portion of the total body pool that is stored in cells specialized in storing the metal and processing heme (e.g., hepatocytes and macrophages). In healthy individuals, the normal concentrations range between 15 and 300 μg/L, with lower values in children vs. adults, in women vs. men, and in fertile vs. post-menopausal women. Normally, 1 μg/L of serum ferritin corresponds to 8–10 mg of stored iron as a direct proportion. Values comprised between 12 and 15 μg/L indicate a depletion of iron stores. The ferritin measurement is widely available, standardized, and methodologically robust, and is based on colorimetric/fluorescent enzyme-linked immunoassays (ELISA) or on chemiluminescent immunoassays (CLIA) ran on automated analyzers [51]. The serum is the best matrix for a proper ferritin measurement, although plasma is also suitable depending on the analytical method.*Iron supply discrepancies.* In the second stage of deficient erythropoiesis, the decreased rate is ascribed to inadequate iron supply to the bone marrow. While Hb has still normal values (>115g/L), ferritin further reduces (<20 μg/L) together with Tf saturation (<16%). Contrariwise, there is an increase of the sTfR (>1.75 mg/dL) [36]. When the functional requirements are not met by dietary absorption or storage release, serum iron (i.e., the amount of Fe^3+^ in the blood bound to Tf) decreases while Tf increases. Because of this liaison, three assays that measure the potential of iron supply are generally performed concomitantly, being the serum iron, the Tf concentration (reported as the quantity of iron that can be bound to Tf = total iron binding capacity, TIBC), and the percentage of Tf saturation (serum iron × 100/TIBC) [52]. Serum iron can be measured by either colorimetric assays (most used) or atomic absorption spectrophotometry [53]. The concentration of serum transferrin can be measured by immunologic methods (direct) or throughout the determination of TIBC, whose assay is identical to the serum iron assay, but applies an additional step (saturation of iron-binding sites of the transferrin molecule with excess iron) followed by the removal of the unbound iron. Several analyzers measure also the unsaturated iron binding capacity (UIBC), with TIBC being subsequently calculated by summing UIBC to serum iron [54]. Serum iron, TIBC, and transferrin saturation are indexes of an adequate iron supply, but their utility as screening tools for iron deficiency is limited by several factors, such as the circadian rhythm (e.g., morning peak of serum iron and Tf saturation), diet, and oral contraceptive use [55]. Nevertheless, a Tf saturation < 16% is known to reflect a suboptimal iron supply for the proper erythrocyte development [52]. Normal values of serum iron range between 65 μg/dL to 170 μg/dL in adult males and 50 μg/dL to 170 μg/dL in adult females. TIBC and Tf saturation normal ranges are 250–450 μg/dL and 20–60%, respectively, in both adult males and females [48]. The serum is the best sample matrix, but also heparin-plasma may be used, whilst EDTA- and citrate-plasma are unsuitable due to the chelating properties of these anticoagulants. Cellular ion demands [56], the erythroid proliferation rate [57], and the stainable bone marrow iron [58] are known to be linked to the concentrations of the soluble form of the serine protease-cleaved membrane receptor (sTfR) that circulates in plasma bound to Tf. Several lifestyle factors affect sTfR, such as smoking, alcoholic drinking, sedentary behaviors, and hypernutrition [36]. Latex-enhanced immunoassays (nephelometry and turbidimetry) and the more recent immunofluorometric assays have been implemented to evaluate sTfR. However, the usefulness of commercial kits is limited by the poor comparability between different tools, comprising the calibrators (free vs. transferrin-complexed form, tissue origin), the antibodies (monoclonal vs. polyclonal), and reporting units (mg/L vs. nmol/L) [59]. This lack of commutability together with the relatively high cost of reagents are some of the reasons why sTfR measurements have not been widely adopted in clinical practice. Normal range of sTfR are 0.30–1.75 mg/dL. The serum is the best matrix and it should be separated within 8 h from blood drawings in order to get reliable results [48]. Of note, the sTfR/serum ferritin ratio may be more reliable than each parameter alone for the identification of iron deficiency [60].*Iron-deficient anemia.* The third stage of iron-deficient anemia is characterized by a reduction of both Hb concentrations and RBCs below-optimal levels (i.e., functional iron deficiency = iron supply is inadequate to meet the requirements for erythropoiesis). In the absence of ongoing inflammatory processes, the biochemical features are low ferritin (<12 μg/L), Tf saturation (<16%), and Hb (<115 g/L), but high sTfR (>1.75 mg/dL) and RBC protoporphyrin (>80 μg/dL). During the ferrochelatase-dependent insertion of ferrous iron in the proto-porphyrin ring, zinc can alternatively be incorporated to form zinc protoporphyrin, which is normally found in trace amounts [61]. In the early stages of reduced erythropoiesis, erythrocyte zinc protoporphyrin progressively rises, thereby providing to be a useful parameter for detecting uncomplicated functional iron deficiency. Importantly, its measure represents the average iron availability for erythropoiesis during the preceding 3–4 months since they are established during erythrocyte maturation and remain unaltered for the mature RBC lifespan. This value can be measured directly by hematofluorometer (porphyrins fluoresce in the red wavelengths when opportunely excited) or after extraction of the zinc moiety using ethyl acetate and hydrochloric acid. In this latter case, the zinc-free erythrocyte protoporphyrin is measured by conventional fluorometry. Values > 150 μmol/mol heme are highly suggestive of iron deficiency [62]. Although RBCs represent the largest functional compartment, their associated indices are not representative of the individual’s martial status. Hb concentration is usually relevant for assessing the degree of severity of iron deficiency, but its sensitivity is low because of the rather inconsistent variations between healthy and iron-deficient individuals. In addition, the specificity of this test is poor. The packed cell volume (hematocrit, Hct), although widely used in the past, does not provide any additional information to Hb concentration. Altered RBC indices, meaning a reduction of mean corpuscular volume (MCV), a reduction of mean corpuscular hemoglobin (MCH), and an increase of red blood cell distribution width (RDW), are usually a feature of iron-deficient erythropoiesis, but they lack specificity [36,48]. Conversely, modern analyzers can measure reticulocyte and hypochromic cell parameters, such as the reticulocyte Hb and the proportion of hypochromic erythrocytes, which may be useful for a proper assessment of anemia in chronic conditions characterized by a generalized inflammatory state. For instance, the biochemical feature of functional iron deficiency in chronic heart failure can show normal Hb values [63] and higher cut-off limits for both Tf saturation (<20%) and ferritin (<300 μg/L) [64]. Heightened values of ferritin may be also found in chronic kidney disease patients, where the concomitant proteinuria, low-iron diet, and inflammation expose them to veiled iron-deficient conditions [65]. The proportion of hypochromic erythrocytes with the reticulocyte Hb count could be used in these cases though, also for predicting the responsiveness to iron therapy [66].

## 5. The Present of Iron Medicine

### 5.1. Iron Foods

The American National Heart, Lung, and Blood Institute (NHLBI) defines healthy eating changes as first-line treatments for mild to moderate iron-deficiency anemia [67]. Male adults and postmenopausal women should consume 10–11 mg/day of iron, with ranges adjusting according to physiological (e.g., post-menarche women requires 20 mg/day of iron), dietary (e.g., highest bioavailability is for high meat/fish diets), or environmental factors (e.g., the infected host requires increased iron needs). For instance, iron requirements in conditions of lowest bioavailability can be set at 27.4 mg/day for men and 58.8 mg/day for women [68]. Dietary intakes should guarantee the replenishment of daily basal losses, estimated to be around 0.95–1.00 mg through enterocyte exfoliation, small bleeding events, epithelial desquamation, sweat) [69]. Heme-iron from Hb and myoglobin is efficiently absorbed (15–40% of intake) and accounts for 40% of total iron in animal foods whereas non-heme iron represents the totality of iron present in plant foods [70]. Despite the amount of iron in plants greatly surpassing the content in animal sources (see Table 1), it is much lesser absorbed (1–15% of intake) [71]. Overall, the most recognized animal source of iron is the liver from *Bovidae*, such as the calf, but also the one from pigs, sheep, horses, and ducks. Other animal sources with great iron amounts are the kidney, the brewer’s yeast, meats, yolk of chicken eggs, and fishes, such as herrings [72,73].

### 5.2. Dietary Patterns

Many absorption influencers other than the nature of the metal itself influence the bioavailability of iron on a daily base. In Western diets, the bioavailability of iron is 14–18% because of the highest intakes of meats, fishes, and sources of L-ascorbic acids. For instance, highest contents of L-ascorbic acid can be found in some fruits, such as red raspberry (198 mg/100 g), kiwi (141 mg/100 g), lemon (129 mg/100 g), and orange (50 mg/100 g), but also in many other sources like peppers (584 mg/100 g), cabbages (348 mg/100 g), onion or garlic (183 mg/100 g) and veal and other mammal liver (31 mg/100 g) [72]. Of note, vitamin C content in plants fluctuates according to the subspecies, variety, cultivar, ecotype, chemotype, soil, nourishment, geographical location, environmental impact, season of growth and harvest, climate, agricultural practices [74]. Plant-based diets have an iron absorption around 5–12% [75], mainly because of the prevalence of its ferric form. The higher the vegetable intakes the greater the extent of potential interferences of proteins involved in iron homeostasis. Conversely, the higher the variability of food quality the higher the probability that the requirements of important co-factors for hematopoiesis are met, such as those of vitamin B9 and B12. The daily needs of 330 μg of folates equivalents (folates = 1:1, folic acid = 1:1.7), the 5 μg of cobalamin, but also the 650 μg of vitamin A equivalents [76] (retinols = 1:1, pro-vitamin A carotenoids = 1:6 of β-carotene and 1:12 of other carotenoids) should be advised [77]. The highest contents of cobalamin (150–20 μg/100 g) are found in beef and horse liver, clam, mussel, crab, and octopus. Highest contents (500–3000 μg/100 g) of folate are found in beef liver, wheat sprouts, dried beans, brewer’s yeast, egg yolk, soy, peanut, oregano, nettle, asparagus. Highest contents (18,000–500 μg/100 g) of retinol are found in cod liver oil, liver, eel, butter, chicken egg, pecorino cheese, caviar. Highest contents (36,000–3000 μg/100 g) of pro-vitamin A carotenoids are found in paprika, parsley, carrots, basil, sweet potatoes, cabbage, red pepper, yellow pumpkin, mango, radicchio [72].

### 5.3. Fortified Foods

A specific compound can be added to a food matrix through manual means during food processing (i.e., fortification) or earlier during plant growth (i.e., biofortification). Concerning food fortification, the metal was first added during food processing to increase the population intake, but technical and sensory problems occurred, such as rancidity and color changes of the final product. Foods with long shelf lives are therefore fortified with the more stable carbonyl or electrolytic iron powders other than the more soluble ferrous sulphate [78]. These microspheres of pure iron are also known to have high bioavailability [79]. Partial resolutions were obtained when either a micronized form of ferric pyrophosphate or the encapsulated ferrous fumarate have been used to fortify iodized table salt [80], thus keeping it away from uncontrolled redox reactions, or after investigating more stable and effective formula (e.g., iron-casein complex) to be incorporated in foods [81]. Concerning biofortification, advances in crop sustainment valorized the plant’s need for iron to obtain iron-enriched foods, mainly through innovative agronomic practices and even modern genetic adjustments [82]. In fact, plants have basic and adaptation mechanisms to incorporate the metal at the root-soil interface (see Section 2) to avoid iron-deficiency symptoms, such as stunted root growth and interveinal chlorosis of young leaves. Biofortification techniques focus on promoting iron incorporation to allow the obtainment of iron-fortified foods [83], but they also aim at obtaining the greatest bioavailability [84]. Despite being a promising agriculture-based approach, there is still limited evidence regarding the clinical efficacy of these biofortified foods to improve nutritional status [85].

### 5.4. Hands-On Approaches

Anemic conditions are prevalent in rural populations, where nutrition can be scarce or limited to certain categories of food sources (i.e., lack of food security). In these conditions, multifaceted options are applied to avoid dire consequences in poor individuals. Anemia in early life can be counteracted through delayed cord clamping [86] and the use of a small, lightweight fish-shaped iron ingot to be placed in cooking pots, which was shown to leach the metal into food providing an enriched iron source [87]. In these areas, lead—a well-known negative effector on iron absorption—is used not only to make cooking pots, but it is also present at high levels in ground soils [88], with contaminations arising from tube well water procurement. Other interventions may act at neutralizing the negative effectors that worsen the iron status, being infective agents, inflammatory statuses, or lead contamination. In helminth or malaria endemic zones, the infection with hookworm or *Plasmodium* is known to be associated with gastrointestinal bleedings [89] and low-grade inflammation [90], respectively. The handling of helminth infections and the integration of anti-malaria treatment are associated with greater iron homeostasis [86] and should be advised before increasing oral iron intake in order to avoid counterproductive effects (e.g., the feeding of the infective agent at the expense of the host) [91]. The replacement of lead cooking pot should be also targeted. Treating foods with enzymes that degrade other absorption disruptors, such as phytic acid [92], or overcooking plant foods are other pragmatic options that help increase iron bioavailability, but collateral depletions of sensitive nutrients can occur.

### 5.5. Dietary Supplements

People living in poverty may not have access to high-iron foods and pragmatic hands-on approaches are not always implementable in rural areas. Micronutrient powders (i.e., sachets containing dry micronutrient powder to be added to food) may nevertheless improve the martial status of vulnerable individuals, especially infants and young children, as part of the home fortification interventions for low-to-middle income countries supported by UNICEF and CDC [93]. In developed counties, diverse oral iron formulas are also available to sustain older patients before and after orthopedic surgery when hemorrhagic conditions arise. The bioavailability, efficacy, and safety of the iron formula often depend upon the user’s health. Even though micronutrients powders (i.e., coated ferrous fumarate) proved to be effective for reducing anemia rates [94,95], their use should be carefully tailored because of the uncertain safety of increasing oral iron in infants with immature gut [96] or in areas with endemic infective agents [97]. Of note, comparable bioavailability to ferrous fumarate has been observed for ferrous sulphate [98], the latter still remaining among the most used. Concerning other fragile individuals, a multipart formula may be used, such as a sucrosomial matrix of ferric pyrophosphate for older adults undergoing orthopedic surgery [99] or a polysaccharide-iron complex of ferric polymaltose for pregnant women with iron-deficiency anemia [100]. These pharmaceuticals may be preferred because the metal is prevented to get in contact with enterocytes, thus possibly reducing local inflammation [101]. The other ingredients of the formula should be promptly mixed to obtain synergic effects, such as the case of iron plus L-ascorbic acid, but perhaps more satisfying results may be obtained if coupled also with vitamin B12, B9, and vitamin A. Nevertheless, the massive accessibility of dietary supplements expose patients to side effects or misuses, also because of their ease of administration and relatively low costs [102,103]. Even health professionals often lack of intelligent interventions as oral treatments are non-adapted to age, sex, timing either within the same day or through alternative days, or lifestyle behaviors, such as inhabitation altitude or smoking habits.

### 5.6. Parenteral Routes: Transfusions and Injections

Detailed indications regarding first blood transfusions date back to the 17th century, when blood was meant to flow from the artery of a youth into the artery of an aged man. Indirect records reported that even red wine was injected into the veins of hunting dogs to boost their performances [104]. Today, both autologous and allogeneic blood transfusions are considered a valuable iron source, but the latter certainly expose institutions to high costs. The prolonged deposit repletion time and impaired absorption render oral supplements vain for patients who require a rapid iron replacement, such as those suffering from heart or kidney disease [63,105]. Injections of iron-carbohydrate complexes can be the ideal approach, delivering the metal directly into the bloodstream to guarantee the fast replenishment of deposits. The carbohydrate shell helps to isolate the metal from blood components until the complex enters the macrophages of the spleen, the liver, and the bone marrow to be either stored or used. A single dose of intravenous iron may be sufficient to optimize the martial status in fragile individuals, such as older adults who are scheduled for elective orthopedic surgery, whereas oral supplements may require daily administrations for weeks [99]. Diverse intravenous iron formulas are available, with differences in unit size, nature of the carbohydrate shell (e.g., dextran, sucrose, gluconate, maltose, sorbitol), surface charge, iron form (Fe^2+^ or Fe^3+^) and content [106]. The dose of iron to be administered through parenteral routes can be calculated based on body weight and Hb levels [107], whereas the personalization of oral therapy is often missing, probably due to the perception that the vein infusion is riskier. Indeed, most of the current evidence on safety issues comes from poorly-designed small-scale trials with short follow-ups, possibly concealing long term risks of iron overload or tissue damage, especially for patients undergoing injections with concomitant high ferritin [108]. Despite this widespread mistrust, most of the formulations are safe and supported by a positive benefit-risk ratio when using tailored dosing and monitoring [109,110], and appears to be more indicated than oral preparations also in conditions of gastrointestinal inflammation or when compliance to oral therapy is dubious. Nevertheless, the diversities in the costs for production, transport, storage, handling (e.g., dilution, contamination risk, in-use stability), and health care assistance render the intravenous preparations not usually considered the first choice of treatment [106,111].

## 6. Conclusions

Iron is a transition metal that had accompanied the evolution of the *Homo* genus throughout its entire evolutionary course. It was first a protagonist in ancient mythology in the form of Mars and “Her Desher”, then in folk medicine in the form of anti-weakness medication, and today it is associated with innumerable health and disease conditions. Iron knowledge has progressively increased and its importance for human health has now very different connotations than in the past. In Figure 1, we have summarized all concepts related to iron dietary sources, to a subject’s martial status, and to anatomical sites that are relevant for the metal homeostasis. Today, anemia conditions affect approximately a third of the world’s population, with great repercussions from before human fertilization [112], through childhood [113], and aging [114]. Natural heavy and acid rains progressively contribute to washing away precious minerals from the soil whose acid-buffering capacity is increasingly disturbed. Both single [115,116] and multimodal [117] nutritional interventions have been investigated in community-based or clinical settings to protect fragile individuals from various vitamin or mineral deficits, but iron insufficiency still seems to persist as quite a perplexing and underdiagnosed issue even in developed countries [118]. Even after the diagnosis, either the lack of treatment tailoring or the poor compliance of the patient prevent this condition to be cured [119]. Wellness features like obesity, regular blood donations, or even ethical choices, which lead to consuming strict plant-based diets or contrariwise the most desirable white (low-iron) meat obtained from milk-fed anemic veal calves, are just some of the causes attributable to iron deficiency syndromes. The older the body the more it is exposed to malabsorption syndromes, intestinal bleeding, urinary iron loss, cancer, and polypharmacotherapies [120]. Pragmatic solutions that aim at optimizing the martial status at the population level would be required in the near future, with high-iron foods, oral supplements, or intravenous infusions certainly requiring multimodal and tailored interventions to local conditions and populations of interest.

## Figures and Tables

**Figure 1 nutrients-12-01761-f001:**
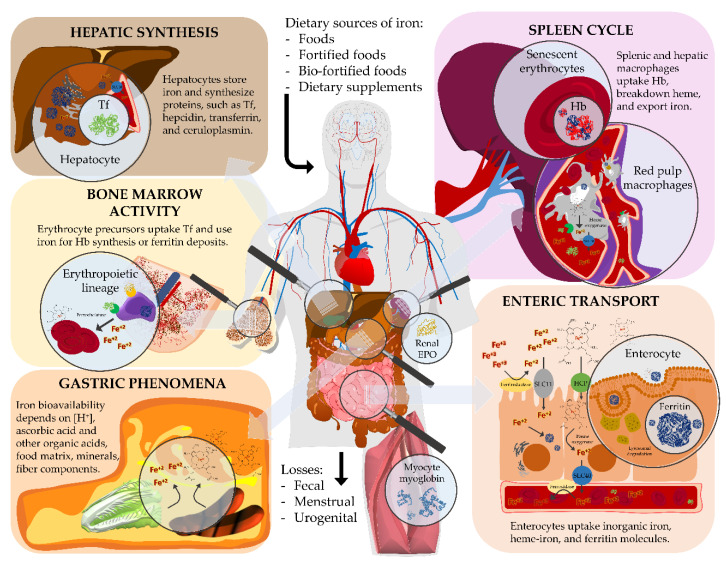
Schematics of the biological pathways of iron with details about major molecules and anatomical sites involved. Dietary iron is absorbed by duodenal enterocytes, circulates in plasma bound to transferrin, and is mainly used to form hemoglobin in newly synthesized red blood cells. Most of body iron is recycled by rep pulp macrophages that engulf senescent erythrocytes and degrade heme to restore circulating transferrin saturation. Iron deposits are mainly at the level of intestines and liver, whereas bone marrow and spleen represent the initial and ultimate step of iron metabolism, respectively. Hb = hemoglobin; Tf = transferrin; SLC11 = proton-coupled metal ion transporter; HCP = heme carrier protein; SLC40 = basolateral transporter ferroportin; EPO = erythropoietin.

**Table 1 nutrients-12-01761-t001:** Highest natural dietary sources of iron in decreasing order.

Dietary Source	Average Contents of Iron, mg/100g Step
	(Daily Needs for Adults: 10–11 mg)
**Animal foods**	
Veal and other mammal liver, raw	20
Yolk of chicken eggs, raw	5
Fishes, raw	5
Meats (veal, beef), raw	4
Milk (cow), whole	0.2
**Plant foods**	
Common oregano, dried	18
Bitter cocoa, powder	14.3
Arabica coffee, powder	12
Dried pulses (lentils, beans), dried	9
Wheat bran, soy flour, dried	8
Walnuts, almonds, pistachios, dried	7
Edible mushrooms, raw	1–2
Red wine	0.9–1.1

Dietary supplements or enriched sources have been excluded from the list. Average amounts of commonly consumed foods have been reported from FooDB v.1.0 (http://foodb.ca/), Dr. Duke’s Phytochemical and Ethnobotanical Databases v.1.9.12.6-Beta (https://phytochem.nal.usda.gov/), and the Italian food databases BDA v.2015 (http://bda-ieo.it/) and [72]. For the same weight, spices, herbs, and vegetables contain large amounts of iron compared to animal foods. However, these plants contain inorganic iron, which is poorly absorbed, and are consumed in very small quantities mainly as flavor boosters.

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
