# Peer review of "The Central Role of Iron in Human Nutrition: From Folk to Contemporary Medicine"

_nutrients, 2020, doi:10.3390/nu12061761_

Round 1
Reviewer 1 Report
Ln 40 to 45: I am not (and I believe the authors neither) an historian and I am not able to challenge this (interesting) information. However, I would prefer a more “scientific” reference than a TV video with no sources. Please, identify an academic source ( book, article, textbook, …) to support your statements
Ln 45: change paragraph
1st section. Please add a few words about the oxido-reductive balance of iron and its potential chemical and biochemical roles, as it will introduce what is mentioned in the following sections and ease the understanding ( eg ln 66 and 67, and others)
Section 2 (ln 63 to 87)would strongly benefit from some re-writing and development. The iron metabolism in plants is of limited interest, while developments on hemoglobin would be warranted, without forgetting myoglobin and with more details about anemia, which is not the same thing as iron deficiency. There is extensive literature about this that can easily inform this point, which is not treated here and needs to be.
Ln 95-98. Although it’s a nice smile, the mention of “Asterix” has nothing to do in such a review. Please remove.
Ln 107-110: would you have a reference for this interesting information?
Section 4.1 ( ln 112 to 153) is interesting, but hard to follow. Please, introduce sub-sections, clarify overall text and improve English language. A schematized vision may be relevant. Provide references for statements made in tln 129 to 145, which lack support. Develop the role of hepcidin and its function in iron homeostasis. Indeed, it is the master regulator of iron : it controls its dietary absorption, storage, and tissue distribution.
Section 4.2
Ln 158-159. Please explain the link between ulcers, H pylori & iron absorption ( develop on pH and cross reference with above section)
Ln 164: the chemical formula of vit C is not needed. Please, remove.
Ln 166: the sentence “Despite only a small fraction of ingested vitamin C being absorbed” is incorrect. Please remove.
Section 4.3. is oversimplistic and leads to think that diagnosing iron deficiency is an easy job, which is far from reality. There are hundreds of reviews and scientific papers about this and this section should be totally re-written. Do not forget the role of inflammation which makes difficult to interpret iron markers. (ex of ref is Suchdev PS, Namaste SM, Aaron GJ et al. (2016) Overview of the Biomarkers Reflecting Inflammation and Nutritional Determinants of Anemia (BRINDA) Project. Adv Nutr 7, 349-356., or WHO (2004) Joint World Health Organization/Centers for Disease Control and Technical. Consultation on the Assessment of Iron Status at the Population Level. Assessing the Iron Status of Populations. 2nd ed. Geneva, but many other exist)
Table 1 provide information which is not put into perspective, with no discussion, and no ranking of preferred, most valid methods, and in which cases. Why blood Hb does not appear?
Section 5.
A few words about the current status of iron deficiency are needed. Publications such as Stevens GA, Finucane MM, De-Regil LM et al. (2013) Global, regional, and national trends in haemoglobin concentration and prevalence of total and severe anaemia in children and pregnant and non-pregnant women for 1995-2011: a systematic analysis of population-representative data. Lancet Glob Health 1, e16-25. could be used and quoted. This would help showing that iron deficiency exist everywhere, but is probably more frequent and worryng in young children and (pregnant) women of developing countries and/or ow income areas.
Also a few wods should be said about the dietary recommendations, and distinguish child-bearing age women from men ( not done in ln200 which addresses this later); mention also that Iron recommendation sometimes uses different % of absorption to take into account diets with poor animal foods ( see WHO FAO documentation)
In general (ie outside specific pathological conditions), iron should be delieverd through diet and thus it would be more logical to start this section by foods rather than by dietary supplements.
The section on dietary supplements is written only under the angle of those sold in developed countries and the question of supplements provided to lower income population by large programme, usually held by national or international bodies should be distinguished and treated. No information either is given about speciation ( ie which form od iron is the most efficient? It is not useful to lits (some of) them without giving any other information. To be deeply revised.
Section on iron foods.
Ln 203. Milk is an animal food. Remove “milk, and its derivatives”
Ln 205. Plant iron is not “less consumed”. Remove
Provide range of % of absorption for iron in animal foods and in plant foods
Table 2. it’s not interesting to provide nearly only example of herbs or spices which are never consumed in 100g amounts. Consider rather the contribution and thus change these with foods less concentrated but conused in high amounts ( pulse, whole grain are good examples, but other should be included)
Table 3 is not useful in an iron-paper. Provide some more info in text and remove it.
Section 5.4. change title for “fortified foods”. These are not “functional foods”.
Discuss about the form of iron used to fortified (see section about dietary supplementation)
Section 5.5. there is no need for a separate section and this information can be split in appropriate other sections of the manuscript. And a major “hands on” tip is missing, which is the treatment of infections and inflammation. Indeed, this is of outmost importance as it has been shown that providing iron to an infected person indeed “feed the infection and not the host”. This should be highlighted ( see for ex Jaeggi T, Kortman GA, Moretti D et al. (2015) Iron fortification adversely affects the gut microbiome, increases pathogen abundance and induces intestinal inflammation in Kenyan infants. Gut 64, 731-742. And Sazawal S, Black RE, Ramsan M et al. (2006) Effects of routine prophylactic supplementation with iron and folic acid on admission to hospital and mortality in preschool children in a high malaria transmission setting: community-based, randomised, placebo-controlled trial. Lancet 367, 133-143.)
Author Response
Response to Reviewers for manuscript nutrients-809963
We would like to thank the Editors and the Reviewer 1 for careful and thorough assessment of our manuscript and for the attentive comments and constructive suggestions. In order to comply with your observations, we had the article read to a native English speaker. As for the star rating from 1-to-5, we believe that our revisions can definitely improve your consideration of our manuscript. We do not believe -even before the revision process- that the contribution of our work to the field was to be judged so low. Our mini-review gives a brief, albeit comprehensive, overview that has rarely been seen in the literature. We appreciated all other remarks and recommendations, and we hope that our revised document addresses all-important issues satisfactorily. Our detailed responses to each suggestion are listed below.
Corrections from Authors to Reviewer 1
- Ln 40 to 45: I am not (and I believe the authors neither) an historian and I am not able to challenge this (interesting) However, I would prefer a more “scientific” reference than a TV video with no sources. Please, identify an academic source ( book, article, textbook, …) to support your statements.
We understand the possible lability of the information we mentioned. However, we believe that it is fair to keep it as very similar to what we said in the article. However, we have added the primary academic source that refers to our topic at line 45: “[2] Dickinson O. The Aegean from Bronze Age to Iron Age: Continuity and Change Between the Twelfth and Eighth Centuries BC: Routledge; 2007”.
- Ln 45: change paragraph
We changed the new paragraph at line 46.
- 1st section. Please add a few words about the oxido-reductive balance of iron and its potential chemical and biochemical roles, as it will introduce what is mentioned in the following sections and ease the understanding ( eg ln 66 and 67, and others)
We rewrote the sentences in order to include more information about iron balance and its roles in biological beings. At line 49: “Still existing as full energy form in the core of supernova remnants, common oxidation states of iron are -2, 0, 2 (Fe+2, ferrous status), 3, and 6, with the ferric status (Fe+3) being the most stable on Earth's aerobic atmosphere and neutral pH environments [3]. Nevertheless, this prevalent form remains unavailable for organisms, which evolved to acquire diverse mechanisms for incorporating the metal, such as the reduction of Fe+3 to Fe+2. The inorganic oxide presents as a solid matter, shiny and silvery, hard and dense, good conductor of heat, often forming colored compounds.”.
- Section 2 (ln 63 to 87) would strongly benefit from some re-writing and development. The iron metabolism in plants is of limited interest, while developments on hemoglobin would be warranted, without forgetting myoglobin and with more details about anemia, which is not the same thing as iron deficiency. There is extensive literature about this that can easily inform this point, which is not treated here and needs to be.
We appreciate the comment and improved the entire section “2. The Martial Status in Humans: Although the evolution of biological beings diverged millennials ago, iron had been incorporated by plants and animals in an exceptionally similar manner in order to exploit its role in respiration. Plants acquired two key mechanisms: strategy I (H+ extrusion to promote Fe+3 reduction) and strategy II (release of specific Fe+3-chelating phytosiderophores and subsequent high-affinity uptake) [6]. In Homo sapiens, the complex regulation of deposits allocation (↑ release if ↑ requirements) and erythrocytes (RBCs) cycle assures the metal to be conserved in complex forms bound to proteins, with the intestinal passage (↑ absorption if ↓ reserves) being the main regulatory point of body iron balance. These mechanisms assure a total body iron of about 2.3 g in women and 3.8 g in men, with almost 60-70% being incorporated in the main circulating protein hemoglobin (Hb), 20% in iron deposits of ferritin, and about 15% in other proteins, such as myoglobin, heme enzymes, transferrin (Tf), and other nonheme compounds [7]. The blood content of the metal refers to a subject’s martial status (from Italian profilo marziale: profilo “profile” and marziale “martial”). This allegory still reminds of physical strength and stamina of the Roman god of war, and it is reasonable given the role of these iron-containing proteins in supplying tissues with oxygen. Despite the evolved biological strategies to incorporate iron from environments, both humans and plants commonly suffer from iron deficiency syndromes [8], which refer to the most common form of “anemia” (from Greek ἀναιμá˝·α: ἀν- “without” and -αἷμα “blood”) that is known to affect a third of the world's population. Of note, children and pregnant women of the poorest regions of the world represent the 55% of all anemia cases [9], which derive from the coexistence of physiological increased needs in conditions of both low bioavailability (e.g., cereal-based diet) and other casual factors, such as poverty, hookworm infections, and schistosomiasis [10]. Anemia was already acknowledged in the past when the lack of energy that a soldier could feel before the war could make a difference between victory and defeat. The hemorrhagic anemia often happened after battles and now easily occurs after surgeries. Nowadays, it is known that anemia is ascribed to different conditions, possibly mirroring a dysfunction of hematopoietic organs, hereditary diseases (e.g., sickle cell disease), or secondary conditions (e.g., vitamin B12 deficiency in pernicious anemia).”. Of note, we preferred to include some references about anemia types to the following section “4.3.1. Understanting the iron deficiency” in order to go not too far on topics that will be discussed later.
- Ln 95-98. Although it’s a nice smile, the mention of “Asterix” has nothing to do in such a review. Please remove.
We agree with you that it is a nice smile. We were undecided whether to insert the reference during the writing process and we made an attempt to see if it could be of interest to the reader. We believe that it has something to do with our review because one of the ingredients of the magic potion is said to be the watercress (which we mention in the article a few lines above). However, we understand the little relevance, and therefore we have removed the reference from line 98.
- Ln 107-110: would you have a reference for this interesting information?
Concerning the phrase “They believe that this preparation could sustain the recover from illness and tiredness, which is indeed often associated to low blood iron”, we do not have a reference. All authors are Italian and have direct experience with grandmothers from southern Italy, so it is a statement supported by our personal experience.
- Section 4.1 ( ln 112 to 153) is interesting, but hard to follow. Please, introduce sub-sections, clarify overall text and improve English language. A schematized vision may be relevant. Provide references for statements made in tln 129 to 145, which lack support. Develop the role of hepcidin and its function in iron homeostasis. Indeed, it is the master regulator of iron : it controls its dietary absorption, storage, and tissue distribution.
We thank the Reviewer for the constructive comment. It should be said that we have condensed the main metabolic steps of iron without going too long, not being a comprehensive review on iron metabolism. We agree with the division into sections and with the need of exhausting the more the central role of hepcidin. We added some relevant references. At line 1137 we revised the section 4.1 Overview of Iron Metabolism.
“4.1.1. Gastric processing
Upon ingestion, food is mixed with gastric juice to obtain proper solubilization of different micronutrients. In particular, iron needs to be reduced and prevented to form insoluble complexes upon chelation with low molecular weight substances. Several positive or negative reactions can occur at the level of the stomach (see 4.2. Absorption Influencers) that can prevent the appropriate solubilization of the inorganic iron, thereby influencing its bioavailability upon entrance of the proximal small intestine [15]. Iron absorption mostly occurs at the level of the duodenum, close to the pyloric proximity in order to exploit the residual acidity before pH buffering.
4.1.2. Intestinal passage
The ferrireductase duodenal cytochrome b (DCYTB) helps to keep iron reduced at the absorption site, thus allowing the internalization through the divalent metal transporter (DMT1) [16], which is part of the family of proton-coupled metal ion transporters (SLC11A2). Concerning heme-iron, it is not yet clear how it can be internalized into the enterocyte [17]. The low-affinity heme carrier protein (HCP) has been proposed to have a role, with the metal being subsequently freed from the porphyrin ring by a heme oxygenase [18]. On the apical border, also dietary ferritin may be absorbed through endocytosis and then subjected to lysosomal digestion [19]. If there is positive iron homeostasis, reduced iron can be complexed with apoferritin to form ferritin deposits. If iron is required, the basolateral transporter ferroportin (SLC40A1) exports ferrous iron that is subsequently incorporated into apotransferrin by either the membrane-bound hephestin (copper-dependent ferroxidase, so named from “Hephaestus”, the Greek god of metalworking) or the circulating ceruloplasmin (ferroxidase produced by the liver) [20].
4.1.3. Systemic delivery
Tf binds all iron circulating in plasma and represents the most dynamic compartment, with a turnover rate of about ten times a day that meets the erythropoiesis requirements [21]. The complex Tf-iron interacts with a ubiquitously located receptor and is then internalized through receptor-mediated endocytosis [22]. The subsequent acidification of the vesicle lumen by hydrogen proton pumps allows the offloading of iron-bound Tf and the entry in two different pathways: a recycling pathway, which implies recycling of Tf back to the plasma membrane for iron reloading, and the endosomal degradation pathway, which ends with the release of iron from the endosome thanks to SLC11A2 [23]. Iron can then be sequestered within the deposits of ferritin if it is not required for immediate metabolic functions, such as the synthesis of heme or iron-sulphur clusters [24]. The post-transcriptional regulation of the iron-regulatory protein-1 (IRP1) guarantees the cellular iron homeostasis, which may be inferred from the concentration of circulating ferritin as it is normally secreted by cells in quantities proportional to intracellular deposits. The lesser prevalent hemosiderin is another iron storage complex that less easily releases the metal upon increased requirements [24].
4.1.4. Physiological roles
Hematopoietic tissues incorporate most of the blood iron, whereas other tissues, such as myocytes, internalize smaller quantities. In fact, erythrocyte precursors in the bone marrow of vertebrae, sternum, and ribs, highly express Tf receptors together with that of the kidney erythropoietin (EPO), thereby boosting the differentiation of cells that are part of the erythropoietic lineage during hypoxic conditions [17]. Of note, effective erythropoiesis requires folate and cobalamin to sustain the pyrimidine synthesis, with the terminal enzyme of the heme biosynthetic pathway (i.e. the ferrochelatase) having a key role in catalyzing the insertion of Fe2+ into the proto-porphyrin ring structure to form the heme molecule [24]. In the reticuloendothelial system of the human spleen, resident macrophages of the red pulp are in charge of senescent RBCs clearance [25], being capable of metabolizing hemoglobin through proteolysis, heme through heme oxygenase activity, and ferritin through lysosomal degradation. Unless it is not required, the metal exits the macrophages thanks to the SLC40A1, is oxidized by ceruloplasmin and bound to Tf, thus subsequently replenishing most of the Tf iron pool [26].
4.1.5. Homeostatic regulation
Other than the abovementioned IRP system, which mainly controls cellular iron uptake and deposits, there is also a general regulatory system for iron homeostasis. Primarily produced by hepatocytes, hepcidin is the master regulator that coordinates dietary absorption, storage, and tissue distribution [27]. Increased hepcidin reduces the number of exposed SLC11 and SLC40, thus blocking the intestinal passage. Consequently, it affects the release of iron from macrophages and hepatocytes, the latter having a great capability for iron deposition in the ferritin form [28]. Reduced iron entry into the bloodstream results in low Tf saturation and lesser iron to be delivered to tissues that expose Tf receptors. Dysregulation of these mechanisms results in iron disorders. Anemia from chronic disease is known to be associated with overexpression of hepcidin, macrophage iron loading, low blood iron, and reduced erythropoiesis [29]. Conversely, negligible hepcidin expression causes higher iron entry into the bloodstream, high Tf saturation, and excess iron accumulation in vital organs (e.g., hemochromatosis) [30].”.
New/revised references are:
- Anderson GJ, Frazer DM. Current understanding of iron homeostasis. Am J Clin Nutr. 2017;106(Suppl 6):1559S-1566S.
- Kaplan J. Mechanisms of cellular iron acquisition: another iron in the fire. Cell. 2002;111(5):603-606.
- Gulec S, Anderson GJ, Collins JF. Mechanistic and regulatory aspects of intestinal iron absorption. Am J Physiol Gastrointest Liver Physiol. 2014;307(4):G397-409.
- Le Blanc S, Garrick MD, Arredondo M. Heme carrier protein 1 transports heme and is involved in heme-Fe metabolism. Am J Physiol Cell Physiol. 2012;302(12):C1780-1785.
- Kalgaonkar S, Lonnerdal B. Receptor-mediated uptake of ferritin-bound iron by human intestinal Caco-2 cells. J Nutr Biochem. 2009;20(4):304-311.
- Chen H, Attieh ZK, Su T, Syed BA, Gao H, Alaeddine RM, et al. Hephaestin is a ferroxidase that maintains partial activity in sex-linked anemia mice. Blood. 2004;103(10):3933-3939.
- Dautry-Varsat A. Receptor-mediated endocytosis: the intracellular journey of transferrin and its receptor. Biochimie. 1986;68(3):375-381.
- Luck AN, Mason AB. Transferrin-mediated cellular iron delivery. Curr Top Membr. 2012;69:3-35.
- Grant BD, Donaldson JG. Pathways and mechanisms of endocytic recycling. Nat Rev Mol Cell Biol. 2009;10(9):597-608.
- Abbaspour N, Hurrell R, Kelishadi R. Review on iron and its importance for human health. J Res Med Sci. 2014;19(2):164-174.
- Nagelkerke SQ, Bruggeman CW, den Haan JMM, Mul EPJ, van den Berg TK, van Bruggen R, et al. Red pulp macrophages in the human spleen are a distinct cell population with a unique expression of Fc-gamma receptors. Blood Adv. 2018;2(8):941-953.
- Ganz T. Macrophages and systemic iron homeostasis. J Innate Immun. 2012;4(5-6):446-453.
- Nemeth E, Ganz T. Regulation of iron metabolism by hepcidin. Annu Rev Nutr. 2006;26:323-342.
- Nunez MT. Regulatory mechanisms of intestinal iron absorption-uncovering of a fast-response mechanism based on DMT1 and ferroportin endocytosis. Biofactors. 2010;36(2):88-97.
- De Domenico I, Ward DM, Kaplan J. Hepcidin regulation: ironing out the details. J Clin Invest. 2007;117(7):1755-1758.
- Anderson GJ. Mechanisms of iron loading and toxicity. Am J Hematol. 2007;82(12 Suppl):1128-1131.
- Section 4.2 Ln 158-159. Please explain the link between ulcers, H pylori & iron absorption ( develop on pH and cross reference with above section)
We thank the Reviewer for this request of amplification. We added at line 181 “A slowly bleeding from an ulcer that goes unnoticed may cause hemorrhagic anemia whereas chronic gastritis at the level of the body can cause an acid output reduction. Similarly, H. pylori infection leads to a reduction of the levels of L-ascorbic acid in the digestive fluid juice and some strains of this infective agent are even able to compete with the host for binding iron [32]”. At line 185 we cross referenced with above section “In addition, some medications, such as antacids, are known to substantially reduce iron absorption because of the lumen acidity neutralization, which is known to prevent the reduction of inorganic oxides”.
- Ln 164: the chemical formula of vit C is not needed. Please, remove.
We removed it from line 192.
- Ln 166: the sentence “Despite only a small fraction of ingested vitamin C being absorbed” is incorrect. Please remove.
We removed it from line 193.
- Section 4.3. is oversimplistic and leads to think that diagnosing iron deficiency is an easy job, which is far from reality. There are hundreds of reviews and scientific papers about this and this section should be totally re-written. Do not forget the role of inflammation which makes difficult to interpret iron markers. (ex of ref is Suchdev PS, Namaste SM, Aaron GJ et al. (2016) Overview of the Biomarkers Reflecting Inflammation and Nutritional Determinants of Anemia (BRINDA) Project. Adv Nutr 7, 349-356., or WHO (2004) Joint World Health Organization/Centers for Disease Control and Technical. Consultation on the Assessment of Iron Status at the Population Level. Assessing the Iron Status of Populations. 2nd ed. Geneva, but many other exist). Table 1 provide information which is not put into perspective, with no discussion, and no ranking of preferred, most valid methods, and in which cases. Why blood Hb does not appear?
We agree with this Reviewer's comment although the shortness of this paragraph was simply due to the choice of the authors to focus on iron in nutrition. However, taking a cue from this criticism, we have completely rewritten this paragraph, by taking into consideration all the suggestions made by the Reviewer as regard both the implementation of the information about the diagnostic indexes (based on, but not limited to, the literature suggested) and by discussing the role of inflammation. In order to avoid any redundancy of the information, as well as based on the reorganization of the information in the text, we decided to remove table 1.
At line 198: “4.3.1. Understanting the iron deficiency
Anemia from iron deficiency is the most common anemia type [35] and may derive from inadequate intake (e.g., poor diet quality), malabsorption (e.g., gastritis, celiac disease, gastritis, gastrointestinal resection, iron refractory iron deficiency anemia), increased physiological requirement (e.g., growth, menses, pregnancy), or pathological blood loss (e.g., internal bleedings, menorrhagia, intravascular hemolysis). The nutritional iron deficiency is the most common cause of iron deficiencies and is mainly triggered by increased needs not fully guaranteed by dietary intakes [36]. This condition is eventually associated with a detectable change in different laboratory tests [37, 38]. In 2007, a joint assessment of the WHO and the Centers for Disease Control and Prevention (CDC) indicated ferritin as a primary measure of the martial status at the population level and the soluble Tf receptor (sTfR) as a second promising parameter that warranted continued evaluation [39]. These two biomarkers are useful to categorize the anemia type as both mirror the intracellular iron homeostasis. As abovementioned, small quantities of ferritin are present in the serum reflecting the amounts deposited in cells. Similarly, small amounts of sTfR derive from the extracellular cleavage of the Tf receptor, and increased serum levels mirror negative iron homeostasis [40, 41]. Nevertheless, ferritin is also an acute-phase protein involved in the inflammatory response against pathogens therefore being of limited use during infective and inflammatory conditions, but also in case of liver disease, tumor, hyperthyroidism, and heavy alcohol intake [42]. If not properly assessed, the prevalence of anemia may be underestimated [43], as ferritin increases during inflammatory conditions irrespective of the martial status [44]. Consequently, it has been suggested to rise the cut-off value from 12 to 30 μg/L since an adjustment of ferritin values according to the individual’s inflammatory status has found no consensus yet. The sTfR is less influenced by inflammation, but other acute-phase mechanisms, such as hypoxia or iron-limited erythropoiesis, are known to possibly affect its circulating levels [45]. Regardless of the etiology, frank anemic conditions represent risk factors for bad conditions, especially in fragile individuals [46, 47], and specific diagnostic algorithms are available to categorize the type to properly tailoring the intervention.
4.3.2. The martial status biomarkers during iron deficiency
The depletion of storages, iron-deficient erythropoiesis, and iron-deficient anemia are the increasingly severe consequences that arise upon iron deficiency, with the affection of erythroid cell development and feature being acknowledged by impaired RBCs homeostasis but even intracellular iron-containing proteins [48]. Although the measurement of blood parameters relies on well-established and widely used analytical methods, many concerns persist regarding the pre-analytical phase management and assay comparability/standardization.
- Iron storage depletion. During the first phase of iron depletion, the deposits in the bone marrow, liver, and spleen are becoming exhausted (no stainable bone marrow iron), but no consequences on erythropoiesis are detectable yet. This early depletion is characterized by low ferritin (< 35 μg/L), but normal Hb and other martial status indices [36]. The bone marrow is a major site for iron storage, but all the local metal is used for erythropoiesis, easily impairing RBC generation upon iron depletion at this site. The absence of stainable iron in the bone marrow is the gold standard for iron deficiency diagnosis, but it is used only in certain circumstances due to the invasive nature of the procedure [49]. It is based on the Prussian blue staining of aspirates to detect both hemosiderin in macrophages and iron granules in sideroblasts. The analysis requires an experienced observer and careful attention to detail [50]. The serum fraction of ferritin represents a portion of the total body pool that is stored in cells specialized in storing the metal and processing heme (e.g., hepatocytes and macrophages). In healthy individuals, the normal concentrations range between 15 and 300 μg/L, with lower values in children vs. adults, in women vs. men, and in fertile vs. post-menopausal women. Normally, 1 μg/L of serum ferritin corresponds to 8-10 mg of stored iron as a direct proportion. Values comprised between 12 and 15 μg/L indicate a depletion of iron stores. The ferritin measurement is widely available, standardized, and methodologically robust, and is based on colorimetric/fluorescent enzyme-linked immunoassays (ELISA) or on chemiluminescent immunoassays (CLIA) ran on automated analyzers [51]. The serum is the best matrix for a proper ferritin measurement, although plasma is also suitable depending on the analytical method.
- Iron supply discrepancies. In the second stage of deficient erythropoiesis, the decreased rate is ascribed to inadequate iron supply to the bone marrow. While Hb has still normal values (> 115g/L), ferritin further reduces (< 20 μg/L) together with Tf saturation (< 16%). Contrariwise, there is an increase of the sTfR (> 1.75 mg/dL) [36]. When the functional requirements are not met by dietary absorption or storage release, serum iron (i.e. the amount of Fe+3 in the blood bound to Tf) decreases while Tf increases. Because of this liaison, three assays that measure the potential of iron supply are generally performed concomitantly, being the serum iron, the Tf concentration (reported as the quantity of iron that can be bound to Tf = total iron binding capacity, TIBC), and the percentage of Tf saturation (serum iron × 100/TIBC) [52]. Serum iron can be measured by either colorimetric assays (most used) or atomic absorption spectrophotometry [53]. The concentration of serum transferrin can be measured by immunologic methods (direct) or throughout the determination of TIBC, whose assay is identical to the serum iron assay, but applies an additional step (saturation of iron-binding sites of the transferrin molecule with excess iron) followed by the removal of the unbound iron. Several analyzers measure also the unsaturated iron binding capacity (UIBC), with TIBC being subsequently calculated by summing UIBC to serum iron [54]. Serum iron, TIBC, and transferrin saturation are indexes of an adequate iron supply, but their utility as screening tools for iron deficiency is limited by several factors, such as the circadian rhythm (e.g., morning peak of serum iron and Tf saturation), diet, and oral contraceptive use [55]. Nevertheless, a Tf saturation < 16% is known to reflect a suboptimal iron supply for the proper erythrocyte development [52]. Normal values of serum iron range between 65 μg/dL to 170 μg/dL in adult males and 50 μg/dL to 170 μg/dL in adult females. TIBC and Tf saturation normal ranges are 250-450 μg/dL and 20-60%, respectively, in both adult males and females [48]. The serum is the best sample matrix, but also heparin-plasma may be used, whilst EDTA- and citrate-plasma are unsuitable due to the chelating properties of these anticoagulants. Cellular ion demands [56], the erythroid proliferation rate [57], and the stainable bone marrow iron [58] are known to be linked to the concentrations of the soluble form of the serine protease-cleaved membrane receptor (sTfR) that circulates in plasma bound to Tf. Several lifestyle factors affect sTfR, such as smoking, alcoholic drinking, sedentary behaviors, and hypernutrition [36]. Latex-enhanced immunoassays (nephelometry and turbidimetry) and the more recent immunofluorometric assays have been implemented to evaluate sTfR. However, the usefulness of commercial kits is limited by the poor comparability between different tools, comprising the calibrators (free vs. transferrin-complexed form, tissue origin), the antibodies (monoclonal vs. polyclonal), and reporting units (mg/L vs. nmol/L) [59]. This lack of commutability together with the relatively high cost of reagents are some of the reasons why sTfR measurements have not been widely adopted in clinical practice. Normal range of sTfR are 0.30-1.75 mg/dL. The serum is the best matrix and it should be separated within 8 h from blood drawings in order to get reliable results [48]. Of note, the sTfR/serum ferritin ratio may be more reliable than each parameter alone for the identification of iron deficiency [60].
- Iron-deficient anemia. The third stage of iron-deficient anemia is characterized by a reduction of both Hb concentrations and RBCs below-optimal levels (i.e. functional iron deficiency = iron supply is inadequate to meet the requirements for erythropoiesis). In the absence of ongoing inflammatory processes, the biochemical features are low ferritin (< 12 μg/L), Tf saturation (< 16%), and Hb (< 115 g/L), but high sTfR (> 1.75 mg/dL) and RBC protoporphyrin (> 80 μg/dL). During the ferrochelatase-dependent insertion of ferrous iron in the proto-porphyrin ring, zinc can alternatively be incorporated to form zinc protoporphyrin, which is normally found in trace amounts [61]. In the early stages of reduced erythropoiesis, erythrocyte zinc protoporphyrin progressively rises, thereby providing to be a useful parameter for detecting uncomplicated functional iron deficiency. Importantly, its measure represents the average iron availability for erythropoiesis during the preceding 3-4 months since they are established during erythrocyte maturation and remain unaltered for the mature RBC lifespan. This value can be measured directly by hematofluorometer (porphyrins fluoresce in the red wavelengths when opportunely excited) or after extraction of the zinc moiety using ethyl acetate and hydrochloric acid. In this latter case, the zinc-free erythrocyte protoporphyrin is measured by conventional fluorometry. Values > 150 μmol/mol heme are highly suggestive of iron deficiency [62]. Although RBCs represent the largest functional compartment, their associated indices are not representative of the individual’s martial status. Hb concentration is usually relevant for assessing the degree of severity of iron deficiency, but its sensitivity is low because of the rather inconsistent variations between healthy and iron-deficient individuals. In addition, the specificity of this test is poor. The packed cell volume (hematocrit, Hct), although widely used in the past, does not provide any additional information to Hb concentration. Altered RBC indices, meaning a reduction of mean corpuscular volume (MCV), a reduction of mean corpuscular hemoglobin (MCH), and an increase of red blood cell distribution width (RDW), are usually a feature of iron-deficient erythropoiesis, but they lack specificity [36, 48]. Conversely, modern analyzers can measure reticulocyte and hypochromic cell parameters, such as the reticulocyte Hb and the proportion of hypochromic erythrocytes, which may be useful for a proper assessment of anemia in chronic conditions characterized by a generalized inflammatory state. For instance, the biochemical feature of functional iron deficiency in chronic heart failure can show normal Hb values [63] and higher cut-off limits for both Tf saturation (< 20%) and ferritin (<300 μg/L) [64]. Heightened values of ferritin may be also found in chronic kidney disease patients, where the concomitant proteinuria, low-iron diet, and inflammation expose them to veiled iron-deficient conditions [65]. The proportion of hypochromic erythrocytes with the reticulocyte Hb count could be used in these cases though, also for predicting the responsiveness to iron therapy [66].”
Revised/new references are:
- Suchdev PS, Namaste SM, Aaron GJ, Raiten DJ, Brown KH, Flores-Ayala R, et al. Overview of the Biomarkers Reflecting Inflammation and Nutritional Determinants of Anemia (BRINDA) Project. Adv Nutr. 2016;7(2):349-356.
- Lynch S, Pfeiffer CM, Georgieff MK, Brittenham G, Fairweather-Tait S, Hurrell RF, et al. Biomarkers of Nutrition for Development (BOND)-Iron Review. J Nutr. 2018;148(suppl_1):1001S-1067S.
- WHO. Iron Deficiency Anaemia - Assessment, Prevention and Control. A guide for programme managers. 2001.
- Munoz M, Villar I, Garcia-Erce JA. An update on iron physiology. World J Gastroenterol. 2009;15(37):4617-4626.
- Level JWHOCfDCaPTCotAoISatP. Assessing the iron status of populations: including literature reviews: report of a Joint World Health Organization/Centers for Disease Control and Prevention Technical Consultation on the Assessment of Iron Status at the Population Level. Geneva, Switzerland, 6-8 April 2004; 2007.
- Harms K, Kaiser T. Beyond soluble transferrin receptor: old challenges and new horizons. Best Pract Res Clin Endocrinol Metab. 2015;29(5):799-810.
- Shih YJ, Baynes RD, Hudson BG, Flowers CH, Skikne BS, Cook JD. Serum transferrin receptor is a truncated form of tissue receptor. J Biol Chem. 1990;265(31):19077-19081.
- Worwood M. Indicators of the iron status of populations: ferritin. Reference Manual for Laboratory Considerations – Iron Status Indicators for Population Assessments. Geneva: WHO; 2007.
- Thurnham DI, McCabe LD, Haldar S, Wieringa FT, Northrop-Clewes CA, McCabe GP. Adjusting plasma ferritin concentrations to remove the effects of subclinical inflammation in the assessment of iron deficiency: a meta-analysis. Am J Clin Nutr. 2010;92(3):546-555.
- Tomkins A. Assessing micronutrient status in the presence of inflammation. J Nutr. 2003;133(5 Suppl 2):1649S-1655S.
- Dignass A, Farrag K, Stein J. Limitations of Serum Ferritin in Diagnosing Iron Deficiency in Inflammatory Conditions. Int J Chronic Dis. 2018;2018:9394060.
- Munoz M, Acheson AG, Auerbach M, Besser M, Habler O, Kehlet H, et al. International consensus statement on the peri-operative management of anaemia and iron deficiency. Anaesthesia. 2017;72(2):233-247.
- Briguglio M, Gianola S, Aguirre MFI, Sirtori P, Perazzo P, Pennestri F, et al. Nutritional support for enhanced recovery programs in orthopedics: Future perspectives for implementing clinical practice. Nutr Clin Metab. 2019;33(3):190-198.
- Lombardi G, Lippi G, Banfi G. Iron requirements and iron status of athletes. In: Maughan RJ, editor. Sports Nutrition. Encyclopaedia of Sports Medicine. XIX. Oxford, UK: John Wiley & Sons; 2014. p. 229-241.
- Gale E, Torrance J, Bothwell T. The quantitative estimation of total iron stores in human bone marrow. J Clin Invest. 1963;42:1076-1082.
- Barron BA, Hoyer JD, Tefferi A. A bone marrow report of absent stainable iron is not diagnostic of iron deficiency. Ann Hematol. 2001;80(3):166-169.
- Zimmermann MB. Methods to assess iron and iodine status. Br J Nutr. 2008;99 Suppl 3:S2-9.
- Beard J. Indicators of the iron status of populations: free erythrocyte protoporphyrin and zinc protoporphyrin; serum and plasma iron, total iron binding capacity and transferrin saturation; and serum transferrin receptor. Reference Manual for Laboratory Considerations – Iron Status Indicators for Population Assessments. Geneve: WHO; 2007.
- Hedayati M, Abubaker-Sharif B, Khattab M, Razavi A, Mohammed I, Nejad A, et al. An optimised spectrophotometric assay for convenient and accurate quantitation of intracellular iron from iron oxide nanoparticles. Int J Hyperthermia. 2018;34(4):373-381.
- Adams PC, Reboussin DM, Leiendecker-Foster C, Moses GC, McLaren GD, McLaren CE, et al. Comparison of the unsaturated iron-binding capacity with transferrin saturation as a screening test to detect C282Y homozygotes for hemochromatosis in 101,168 participants in the hemochromatosis and iron overload screening (HEIRS) study. Clin Chem. 2005;51(6):1048-1052.
- Bullock GC, Delehanty LL, Talbot AL, Gonias SL, Tong WH, Rouault TA, et al. Iron control of erythroid development by a novel aconitase-associated regulatory pathway. Blood. 2010;116(1):97-108.
- Skikne BS, Flowers CH, Cook JD. Serum transferrin receptor: a quantitative measure of tissue iron deficiency. Blood. 1990;75(9):1870-1876.
- Kohgo Y, Niitsu Y, Kondo H, Kato J, Tsushima N, Sasaki K, et al. Serum transferrin receptor as a new index of erythropoiesis. Blood. 1987;70(6):1955-1958.
- Chang J, Bird R, Clague A, Carter A. Clinical utility of serum soluble transferrin receptor levels and comparison with bone marrow iron stores as an index for iron-deficient erythropoiesis in a heterogeneous group of patients. Pathology. 2007;39(3):349-353.
- Speeckaert MM, Speeckaert R, Delanghe JR. Biological and clinical aspects of soluble transferrin receptor. Crit Rev Clin Lab Sci. 2010;47(5-6):213-228.
- Skikne BS, Punnonen K, Caldron PH, Bennett MT, Rehu M, Gasior GH, et al. Improved differential diagnosis of anemia of chronic disease and iron deficiency anemia: a prospective multicenter evaluation of soluble transferrin receptor and the sTfR/log ferritin index. Am J Hematol. 2011;86(11):923-927.
- Labbe RF, Vreman HJ, Stevenson DK. Zinc protoporphyrin: A metabolite with a mission. Clin Chem. 1999;45(12):2060-2072.
- Tillyer ML, Tillyer CR. Zinc protoporphyrin assays in patients with alpha and beta thalassaemia trait. J Clin Pathol. 1994;47(3):205-208.
- Mordi IR, Tee A, Lang CC. Iron Therapy in Heart Failure: Ready for Primetime? Card Fail Rev. 2018;4(1):28-32.
- McDonagh T, Macdougall IC. Iron therapy for the treatment of iron deficiency in chronic heart failure: intravenous or oral? Eur J Heart Fail. 2015;17(3):248-262.
- Kalantar-Zadeh K, Kalantar-Zadeh K, Lee GH. The fascinating but deceptive ferritin: to measure it or not to measure it in chronic kidney disease? Clin J Am Soc Nephrol. 2006;1 Suppl 1:S9-18.
- Ratcliffe LE, Thomas W, Glen J, Padhi S, Pordes BA, Wonderling D, et al. Diagnosis and Management of Iron Deficiency in CKD: A Summary of the NICE Guideline Recommendations and Their Rationale. Am J Kidney Dis. 2016;67(4):548-558.
- Section 5. A few words about the current status of iron deficiency are needed. Publications such as Stevens GA, Finucane MM, De-Regil LM et al. (2013) Global, regional, and national trends in haemoglobin concentration and prevalence of total and severe anaemia in children and pregnant and non-pregnant women for 1995-2011: a systematic analysis of population-representative data. Lancet Glob Health 1, e16-25. could be used and quoted. This would help showing that iron deficiency exist everywhere, but is probably more frequent and worryng in young children and (pregnant) women of developing countries and/or ow income areas.
We would like to thank the Reviewer for this important consideration that we support. We believe that this consideration about iron deficiency can be added in the section “2. The Martial Status in Humans”. Indeed, we previously stated at line 77 that “Despite the evolved biological strategies to incorporate iron from environments, both humans and plants commonly suffer from iron deficiency syndromes [8], which refer to the most common form of “anemia” (from Greek ἀναιμá˝·α: ἀν- “without” and -αἷμα “blood”) that is known to affect a third of the world's population.”. We added at line 81 “Of note, children and pregnant women of the poorest regions of the world represent the 55% of all anemia cases [9], which derive from the coexistence of physiological increased needs in conditions of both low bioavailability (e.g., cereal-based diet) and other casual factors, such as poverty, hookworm infections, and schistosomiasis [10]”.
The new/revised references are:
- Gyana R Rout, Sahoo S. Role of iron in plant growth and metabolism. 2015;3:1-24.
- Stevens GA, Finucane MM, De-Regil LM, Paciorek CJ, Flaxman SR, Branca F, et al. Global, regional, and national trends in haemoglobin concentration and prevalence of total and severe anaemia in children and pregnant and non-pregnant women for 1995-2011: a systematic analysis of population-representative data. Lancet Glob Health. 2013;1(1):e16-25.
- Camaschella C. Iron-deficiency anemia. N Engl J Med. 2015;372(19):1832-1843.
- Also a few wods should be said about the dietary recommendations, and distinguish child-bearing age women from men ( not done in ln200 which addresses this later); mention also that Iron recommendation sometimes uses different % of absorption to take into account diets with poor animal foods ( see WHO FAO documentation)
We added suggested information and moved the reference values in this section. At line 327 “Male adults and postmenopausal women should consume 10-11 mg/die of iron, with ranges adjusting according to physiological (e.g., post-menarche women requires 20 mg/die of iron), dietary (e.g., highest bioavailability is for high meat/fish diets), or environmental factors (e.g., the infected host requires increased iron needs). For instance, iron requirements in conditions of lowest bioavailability can be set at 27.4 mg/die for men and 58.8 mg/die for women [68].”
- In general (ie outside specific pathological conditions), iron should be delieverd through diet and thus it would be more logical to start this section by foods rather than by dietary supplements.
We support the fact that iron should be first delivered through diet. In fact, al line 485 we stated that “Pragmatic solutions … with high-iron foods, oral supplements, or intravenous infusions …” being consequent choices. Therefore, we moved “5.1 Iron Foods” at line 325 and “5.5. Dietary Supplements” at line 413.
- The section on dietary supplements is written only under the angle of those sold in developed countries and the question of supplements provided to lower income population by large programme, usually held by national or international bodies should be distinguished and treated. No information either is given about speciation ( ie which form od iron is the most efficient? It is not useful to lits (some of) them without giving any other information. To be deeply revised.
We agree with the comment and are aware of the humanitarian programs (e.g. Improve infant and Young Child Feeding - IYCF) supported by UNICEF and CDC. We therefore added some information about these micronutrient powders to be used for home fortification. In addition, we rewrote the paragraph in order to include more details about different formula efficacy. At line 414 “Whenever conceivable, it is preferred to resort to dietary supplements to increase the oral iron intake in rural populations. Other than the pragmatic hands-on approaches, micronutrient powders (i.e. sachets containing dry micronutrient powder to be added to food) may improve the martial status of vulnerable individuals, especially infants and young children, as part of the home fortification interventions for low-to-middle income countries supported by UNICEF and CDC [93]. In developed counties, diverse oral iron formulas are also available to sustain patients before and after surgery when hemorrhagic conditions arise. The bioavailability, efficacy, and safety of the iron formula often depend upon the user’s health. Even though micronutrients powders (i.e. coated ferrous fumarate) proved to be effective for reducing anemia rates [94, 95], their use should be carefully tailored because of the uncertain safety of increasing oral iron in infants with immature gut [96] or in areas with endemic infective agents [97]. Of note, comparable bioavailability to ferrous fumarate has been observed for ferrous sulphate [98], the latter still remaining among the most used. Concerning other fragile individuals, a multipart formula may be used, such as a sucrosomial matrix of ferric pyrophosphate for older adults [99] or a polysaccharide-iron complex of ferric polymaltose for pregnant women [100]. These pharmaceuticals may be preferred because the metal is prevented to get in contact with enterocytes, thus possibly reducing local inflammation [101]”.
Revised references are:
- Jefferds ME, Irizarry L, Timmer A, Tripp K. UNICEF-CDC global assessment of home fortification interventions 2011: current status, new directions, and implications for policy and programmatic guidance. Food Nutr Bull. 2013;34(4):434-443.
- De-Regil LM, Suchdev PS, Vist GE, Walleser S, Pena-Rosas JP. Home fortification of foods with multiple micronutrient powders for health and nutrition in children under two years of age (Review). Evid Based Child Health. 2013;8(1):112-201.
- De-Regil LM, Jefferds MED, Pena-Rosas JP. Point-of-use fortification of foods with micronutrient powders containing iron in children of preschool and school-age. Cochrane Database Syst Rev. 2017;11:CD009666.
- Jaeggi T, Kortman GA, Moretti D, Chassard C, Holding P, Dostal A, et al. Iron fortification adversely affects the gut microbiome, increases pathogen abundance and induces intestinal inflammation in Kenyan infants. Gut. 2015;64(5):731-742.
- Sazawal S, Black RE, Ramsan M, Chwaya HM, Stoltzfus RJ, Dutta A, et al. Effects of routine prophylactic supplementation with iron and folic acid on admission to hospital and mortality in preschool children in a high malaria transmission setting: community-based, randomised, placebo-controlled trial. Lancet. 2006;367(9505):133-143.
- Harrington M, Hotz C, Zeder C, Polvo GO, Villalpando S, Zimmermann MB, et al. A comparison of the bioavailability of ferrous fumarate and ferrous sulfate in non-anemic Mexican women and children consuming a sweetened maize and milk drink. Eur J Clin Nutr. 2011;65(1):20-25.
- Briguglio M, Hrelia S, Malaguti M, De Vecchi E, Lombardi G, Banfi G, et al. Oral Supplementation with Sucrosomial Ferric Pyrophosphate Plus L-Ascorbic Acid to Ameliorate the Martial Status: A Randomized Controlled Trial. Nutrients. 2020;12(2).
- Ortiz R, Toblli JE, Romero JD, Monterrosa B, Frer C, Macagno E, et al. Efficacy and safety of oral iron(III) polymaltose complex versus ferrous sulfate in pregnant women with iron-deficiency anemia: a multicenter, randomized, controlled study. J Matern Fetal Neonatal Med. 2011;24(11):1347-1352.
- Asperti M, Gryzik M, Brilli E, Castagna A, Corbella M, Gottardo R, et al. Sucrosomial((R)) Iron Supplementation in Mice: Effects on Blood Parameters, Hepcidin, and Inflammation. Nutrients. 2018;10(10).
- Section on iron foods. Ln 203. Milk is an animal food. Remove “milk, and its derivatives”
We removed it from line 336.
- Ln 205. Plant iron is not “less consumed”. Remove
We removed it from line 338.
- Provide range of % of absorption for iron in animal foods and in plant foods
Thanks for the important request for clarification. As extensively reported in the manuscript, the content of iron in food (the chemical form of the metal) may be substantially different from its bioavailability (positive or negative effectors) and its absorption rates (healthy status of the individual). The information that you requested is definitely complement of the two other important information that we had already reported in the text. The first (about iron content) is at line 335 “…Heme-iron …accounts for 40% of total iron in animal foods … whereas non-heme iron represents the totality of iron present in plant foods [70]”. The second important information (about iron bioavailability) is at line 351 “In Western diets, the bioavailability of iron is 14-18% because of the highest intakes of meats, fishes, and sources of L-ascorbic acids” and at line 358 “Plant-based diets have an iron absorption around 5-12% [75], mainly because of the prevalence of its ferric form.”. We therefore integrated the information about iron absorption “…Heme-iron from Hb and myoglobin is efficiently absorbed (15-40% of intake) …Despite the amount of iron in plants greatly surpassing the content in animal sources (see Table 1), it is much lesser … absorbed (1-15% of intake) [71]”.
The new reference is:
- Hunt JR. Bioavailability of iron, zinc, and other trace minerals from vegetarian diets. Am J Clin Nutr. 2003;78(3 Suppl):633S-639S.
- Table 2. it’s not interesting to provide nearly only example of herbs or spices which are never consumed in 100g amounts. Consider rather the contribution and thus change these with foods less concentrated but conused in high amounts ( pulse, whole grain are good examples, but other should be included)
We support your comment and we have removed the references for most of spices. We added values for milk, bitter cocoa, mushrooms, and wine. We also added some information about the weighted food.
- Table 3 is not useful in an iron-paper. Provide some more info in text and remove it.
We provided some information in the text and removed the Table 3. At line 352: “…For instance, highest contents of L-ascorbic acid can be found in some fruits, such as red raspberry (198 mg/100g), kiwi (141 mg/100g), lemon (129 mg/100g), and orange (50 mg/100g), but also in many other sources like peppers (584 mg/100g), cabbages (348 mg/100g), onion or garlic (183 mg/100g) and veal and other mammal liver (31 mg/100g) [72]. Of note, vitamin C content in plants fluctuates according to the subspecies, variety, cultivar, ecotype, chemotype, soil, nourishment, geographical location, environmental impact, season of growth and harvest, climate, agricultural practices [74]”.
- Section 5.4. change title for “fortified foods”. These are not “functional foods”.
We changed at line 372 “5.3. Fortified foods”.
- Discuss about the form of iron used to fortified (see section about dietary supplementation)
We added at line 377 “Foods with long shelf lives are therefore fortified with the more stable carbonyl or electrolytic iron powders other than the more soluble ferrous sulphate [78]. These microspheres of pure iron are also known to have high bioavailability [79]. Partial resolutions were obtained when either a micronized form of ferric pyrophosphate or the encapsulated ferrous fumarate have been used to fortify iodized table salt [80], thus keeping it away from uncontrolled redox reactions, or after investigating more stable and effective formula (e.g., iron-casein complex) to be incorporated in foods [81].”.
New/revised references are:
- Hurrell R, Bothwell T, Cook JD, Dary O, Davidsson L, Fairweather-Tait S, et al. The usefulness of elemental iron for cereal flour fortification: a SUSTAIN Task Force report. Sharing United States Technology to Aid in the Improvement of Nutrition. Nutr Rev. 2002;60(12):391-406.
- Lynch SR, Bothwell T, Powders STFoI. A comparison of physical properties, screening procedures and a human efficacy trial for predicting the bioavailability of commercial elemental iron powders used for food fortification. Int J Vitam Nutr Res. 2007;77(2):107-124.
- Andersson M, Thankachan P, Muthayya S, Goud RB, Kurpad AV, Hurrell RF, et al. Dual fortification of salt with iodine and iron: a randomized, double-blind, controlled trial of micronized ferric pyrophosphate and encapsulated ferrous fumarate in southern India. Am J Clin Nutr. 2008;88(5):1378-1387.
- Henare SJ, Nur Singh N, Ellis AM, Moughan PJ, Thompson AK, Walczyk T. Iron bioavailability of a casein-based iron fortificant compared with that of ferrous sulfate in whole milk: a randomized trial with a crossover design in adult women. Am J Clin Nutr. 2019.
- Section 5.5. there is no need for a separate section and this information can be split in appropriate other sections of the manuscript. And a major “hands on” tip is missing, which is the treatment of infections and inflammation. Indeed, this is of outmost importance as it has been shown that providing iron to an infected person indeed “feed the infection and not the host”. This should be highlighted ( see for ex Jaeggi T, Kortman GA, Moretti D et al. (2015) Iron fortification adversely affects the gut microbiome, increases pathogen abundance and induces intestinal inflammation in Kenyan infants. Gut 64, 731-742. And Sazawal S, Black RE, Ramsan M et al. (2006) Effects of routine prophylactic supplementation with iron and folic acid on admission to hospital and mortality in preschool children in a high malaria transmission setting: community-based, randomised, placebo-controlled trial. Lancet 367, 133-143.)
We partially agree with the comment. We had decided to include this section to summarize the “indirect method” to improve the martial status, especially in those conditions in which the use of iron-rich foods or dietary supplementation is not possible. Indeed, these approaches are mainly used in conditions where food security is lacking. The same references that you brought as an example refer to issues of poor infants and high malaria transmission setting. We therefore believe that, for a better logical subdivision of the text, this paragraph has to be maintained. Nevertheless, we have reorganized it to be more understandable and we have inserted the tip on inflammation. About this latter topic, we deeply agree and thank the Reviewer for pointing this out. Our research group is quite accredited on this. We recently published the article “The Malnutritional Status of the Host as a Virulence Factor for New Coronavirus SARS-CoV-2” in which we stated “… it can be assumed that the healthier is the nutritional status of the host … the lower is the virulence … This transitive relation is not necessarily assumable for all pathogens. Concerning parasitic infections, well-nourished subjects may offer a wealthier environment to developing parasites than malnourished individuals”. That is, “feed the infection and not the host” should never happen, but it highly depends upon the pathogen type. Therefore, we have included some important considerations in the manuscript following your comment. In addition, we integrated the beneficial and deleterious role that iron supplementation could have in cases of infection. The referenced that you suggested have been added in the paragraph “5.5. Dietary Supplements” at line 419: “Even though micronutrients powders (i.e. coated ferrous fumarate) proved to be effective for reducing anemia rates [94, 95], their use should be carefully tailored because of the uncertain safety of increasing oral iron in infants with immature gut [96] or in areas with endemic infective agents [97]”. Other suggestions have been used to rewrote the paragraph “5.4. Hands-on approaches”. At line 393 “Anemic conditions are prevalent in rural populations, where nutrition can be scarce or limited to certain categories of food sources (i.e. lack of food security). In these conditions, multifaceted options are applied to avoid dire consequences in poor individuals. Anemia in early life can be counteracted through delayed cord clamping [86] and the use of a small, lightweight fish-shaped iron ingot to be placed in cooking pots, which was shown to leach the metal into food providing an enriched iron source [87]. Other interventions may act at neutralizing the negative effectors that worsen the iron status, being infective agents, inflammatory statuses, or lead contamination. In helminth or malaria endemic zones, the infection with hookworm or Plasmodium is known to be associated with gastrointestinal bleedings [88] and low-grade inflammation [89], respectively. The handling of helminth infections and the integration of anti-malaria treatment are associated with greater iron homeostasis [86] and should be advised before increasing oral iron intake in order to avoid counterproductive effects (e.g., the feeding of the infective agent at the expensive of the host) [90]. In these areas, lead -a well-known negative effector on iron absorption- is used not only to make cooking pots, but it is also present at high levels in ground soils [91], with contaminations arising from tube well water procurement. The replacement of lead cooking pot should be also targeted. Treating foods with enzymes that degrade other absorption disruptors, such as phytic acid [92], or overcooking plant foods are other pragmatic options that help increase iron bioavailability, but collateral depletions of sensitive nutrients can occur.”
Revised references are:
- Campos Ponce M, Polman K, Roos N, Wieringa FT, Berger J, Doak CM. What Approaches are Most Effective at Addressing Micronutrient Deficiency in Children 0-5 Years? A Review of Systematic Reviews. Matern Child Health J. 2019;23(Suppl 1):4-17.
- Charles CV, Dewey CE, Daniell WE, Summerlee AJ. Iron-deficiency anaemia in rural Cambodia: community trial of a novel iron supplementation technique. Eur J Public Health. 2011;21(1):43-48.
- Casey GJ, Montresor A, Cavalli-Sforza LT, Thu H, Phu LB, Tinh TT, et al. Elimination of iron deficiency anemia and soil transmitted helminth infection: evidence from a fifty-four month iron-folic acid and de-worming program. PLoS Negl Trop Dis. 2013;7(4):e2146.
- Cercamondi CI, Egli IM, Ahouandjinou E, Dossa R, Zeder C, Salami L, et al. Afebrile Plasmodium falciparum parasitemia decreases absorption of fortification iron but does not affect systemic iron utilization: a double stable-isotope study in young Beninese women. Am J Clin Nutr. 2010;92(6):1385-1392.
- Briguglio M, Pregliasco FE, Lombardi G, Perazzo P, Banfi G. The Malnutritional Status of the Host as a Virulence Factor for New Coronavirus SARS-CoV-2. Frontiers in Medicine. 2020.
- Tara M. Scrap Metal Pots an Unrecognized Source of Lead Poisoning. Epoch Times. 2017 February 2nd, 2017.
- De-Regil LM, Suchdev PS, Vist GE, Walleser S, Pena-Rosas JP. Home fortification of foods with multiple micronutrient powders for health and nutrition in children under two years of age (Review). Evid Based Child Health. 2013;8(1):112-201.
- De-Regil LM, Jefferds MED, Pena-Rosas JP. Point-of-use fortification of foods with micronutrient powders containing iron in children of preschool and school-age. Cochrane Database Syst Rev. 2017;11:CD009666.
- Jaeggi T, Kortman GA, Moretti D, Chassard C, Holding P, Dostal A, et al. Iron fortification adversely affects the gut microbiome, increases pathogen abundance and induces intestinal inflammation in Kenyan infants. Gut. 2015;64(5):731-742.
- Sazawal S, Black RE, Ramsan M, Chwaya HM, Stoltzfus RJ, Dutta A, et al. Effects of routine prophylactic supplementation with iron and folic acid on admission to hospital and mortality in preschool children in a high malaria transmission setting: community-based, randomised, placebo-controlled trial. Lancet. 2006;367(9505):133-143.

Reviewer 2 Report
Dear Authors,
your paper is interesting and could be helpful for scientists involved in I.D. management and treatment.
I only suggest you to:
1) Review the english (minor spekk checks are required);
2) Better underlined the reason way in 2020 one third of the world population is affected by Iron Deficiency (I.D.) during the chapter 2) Martial Status in Humans;
3) Better underlined the fact that ID is not only a problem in rural or poor population but also in development countries (such as US or Europe);
4) In paragraph 4.3 (Diagnostic of Iron Deficiency) talks about of the new condition of I.D. (Ferritin under 300 Micrg/L and TSat <15%) that, for instance, in Chronic Hearth Failure (C.H.F.) patients significantly worsened the QoL and the survivor times of CHF's patient
5) Better describing the "limits" of iron paremteral route treatment in terms of "cost of illness" (hospitalization), adverse effects and metabolic impairment by administrationg an high quantity of elemental iron in short time;
6) During the CONCLUSION try to analyze the reason why in 2020 more than one third of the world population still suffering from I.D..(bad patient's compliance to domiciliary treatment with oral iron available: how many patients stopped their treatment with Ferrous Sulphate at home "without speaking with their Medical Doctors? Underdiagnosts of I.D. by Medical Doctors, no drugs or Food supplement available for oral routes with high domiciliary compliance; the ancient Ferrous Sulphate is still the "gold standard" for iron repletion treatment). )
Author Response
Response to Reviewers for manuscript nutrients-809963
We would like to thank the Reviewer 2 for the positive comments and constructive suggestions. We appreciated all remarks, and we hope that our revised document addresses all few issues satisfactorily. Our detailed responses to each suggestion are listed below.
Corrections from Authors to Reviewer 2
Dear Authors,
your paper is interesting and could be helpful for scientists involved in I.D. management and treatment.
I only suggest you to:
1) Review the english (minor spekk checks are required);
We would like to thank you for the positive comment. we proceeded to modify some syntax errors.
2) Better underlined the reason way in 2020 one third of the world population is affected by Iron Deficiency (I.D.) during the chapter 2) Martial Status in Humans;
Thank you for the relevant request for clarification. We believe that we have described the various causes that can lead to iron deficiency in several paragraphs (4.2. Absorption Influencers, 5.4. Hands-on approaches, and inflammation in 4.3. Diagnostics of iron deficiency). However, we agree with the Reviewer about the importance of giving an initial overview of the root causes. Regarding this, the article by Camaschella can certainly simplify, as it clearly distinguishes the causes in developing and “more developed” countries. We modified the text accordingly.
At line 81: “Of note, children and pregnant women of the poorest regions of the world represent the 55% of all anemia cases [9], which derive from the coexistence of physiological increased needs in conditions of both low bioavailability (e.g., cereal-based diet) and other casual factors, such as poverty, hookworm infections, and schistosomiasis [10]”
New references are:
- Stevens GA, Finucane MM, De-Regil LM, Paciorek CJ, Flaxman SR, Branca F, et al. Global, regional, and national trends in haemoglobin concentration and prevalence of total and severe anaemia in children and pregnant and non-pregnant women for 1995-2011: a systematic analysis of population-representative data. Lancet Glob Health. 2013;1(1):e16-25.
- Camaschella C. Iron-deficiency anemia. N Engl J Med. 2015;372(19):1832-1843.
As for the causes of ID of developed countries, we think it is more consistent with our text to include them in the discussion (see therefore the answer to comment 6)
3) Better underlined the fact that ID is not only a problem in rural or poor population but also in development countries (such as US or Europe);
We agree with the importance of highlighting also this view. We included this information in the discussion (see the answer to comment 6).
4) In paragraph 4.3 (Diagnostic of Iron Deficiency) talks about of the new condition of I.D. (Ferritin under 300 Micrg/L and TSat <15%) that, for instance, in Chronic Hearth Failure (C.H.F.) patients significantly worsened the QoL and the survivor times of CHF's patient
Thanks for the request for clarification. Based on your comment - but also that of Reviewer 1- we have decided to rewrite the section “4.3. Diagnostics of iron deficiency” entirely in order to further exhaust the subject. Therefore, we proceeded to insert the information that was previously reported in table 1 (which now does not appear in the revised manuscript) in the texts in a more discursive and focused manner on the diagnosis of ID. In the text, we also included some information about your comment on chronic conditions. Precisely with reference to chronic conditions, we have also deepened in the same section the role of inflammation. Your comment is very relevant. In the subsection “iron-deficient anemia we therefore introduced the phenotype of functional iron deficiency that may present increased levels of ferritin because of the generalized inflammation that increased hepcidin (e.g. in CHF). Corrections have been made as follows. At line 198:
“4.3.1. Understanting the iron deficiency
Anemia from iron deficiency is the most common anemia type [35] and may derive from inadequate intake (e.g., poor diet quality), malabsorption (e.g., gastritis, celiac disease, gastritis, gastrointestinal resection, iron refractory iron deficiency anemia), increased physiological requirement (e.g., growth, menses, pregnancy), or pathological blood loss (e.g., internal bleedings, menorrhagia, intravascular hemolysis). The nutritional iron deficiency is the most common cause of iron deficiencies and is mainly triggered by increased needs not fully guaranteed by dietary intakes [36]. This condition is eventually associated with a detectable change in different laboratory tests [37, 38]. In 2007, a joint assessment of the WHO and the Centers for Disease Control and Prevention (CDC) indicated ferritin as a primary measure of the martial status at the population level and the soluble Tf receptor (sTfR) as a second promising parameter that warranted continued evaluation [39]. These two biomarkers are useful to categorize the anemia type as both mirror the intracellular iron homeostasis. As abovementioned, small quantities of ferritin are present in the serum reflecting the amounts deposited in cells. Similarly, small amounts of sTfR derive from the extracellular cleavage of the Tf receptor, and increased serum levels mirror negative iron homeostasis [40, 41]. Nevertheless, ferritin is also an acute-phase protein involved in the inflammatory response against pathogens therefore being of limited use during infective and inflammatory conditions, but also in case of liver disease, tumor, hyperthyroidism, and heavy alcohol intake [42]. If not properly assessed, the prevalence of anemia may be underestimated [43], as ferritin increases during inflammatory conditions irrespective of the martial status [44]. Consequently, it has been suggested to rise the cut-off value from 12 to 30 μg/L since an adjustment of ferritin values according to the individual’s inflammatory status has found no consensus yet. The sTfR is less influenced by inflammation, but other acute-phase mechanisms, such as hypoxia or iron-limited erythropoiesis, are known to possibly affect its circulating levels [45]. Regardless of the etiology, frank anemic conditions represent risk factors for bad conditions, especially in fragile individuals [46, 47], and specific diagnostic algorithms are available to categorize the type to properly tailoring the intervention.
4.3.2. The martial status biomarkers during iron deficiency
The depletion of storages, iron-deficient erythropoiesis, and iron-deficient anemia are the increasingly severe consequences that arise upon iron deficiency, with the affection of erythroid cell development and feature being acknowledged by impaired RBCs homeostasis but even intracellular iron-containing proteins [48]. Although the measurement of blood parameters relies on well-established and widely used analytical methods, many concerns persist regarding the pre-analytical phase management and assay comparability/standardization.
- Iron storage depletion. During the first phase of iron depletion, the deposits in the bone marrow, liver, and spleen are becoming exhausted (no stainable bone marrow iron), but no consequences on erythropoiesis are detectable yet. This early depletion is characterized by low ferritin (< 35 μg/L), but normal Hb and other martial status indices [36]. The bone marrow is a major site for iron storage, but all the local metal is used for erythropoiesis, easily impairing RBC generation upon iron depletion at this site. The absence of stainable iron in the bone marrow is the gold standard for iron deficiency diagnosis, but it is used only in certain circumstances due to the invasive nature of the procedure [49]. It is based on the Prussian blue staining of aspirates to detect both hemosiderin in macrophages and iron granules in sideroblasts. The analysis requires an experienced observer and careful attention to detail [50]. The serum fraction of ferritin represents a portion of the total body pool that is stored in cells specialized in storing the metal and processing heme (e.g., hepatocytes and macrophages). In healthy individuals, the normal concentrations range between 15 and 300 μg/L, with lower values in children vs. adults, in women vs. men, and in fertile vs. post-menopausal women. Normally, 1 μg/L of serum ferritin corresponds to 8-10 mg of stored iron as a direct proportion. Values comprised between 12 and 15 μg/L indicate a depletion of iron stores. The ferritin measurement is widely available, standardized, and methodologically robust, and is based on colorimetric/fluorescent enzyme-linked immunoassays (ELISA) or on chemiluminescent immunoassays (CLIA) ran on automated analyzers [51]. The serum is the best matrix for a proper ferritin measurement, although plasma is also suitable depending on the analytical method.
- Iron supply discrepancies. In the second stage of deficient erythropoiesis, the decreased rate is ascribed to inadequate iron supply to the bone marrow. While Hb has still normal values (> 115g/L), ferritin further reduces (< 20 μg/L) together with Tf saturation (< 16%). Contrariwise, there is an increase of the sTfR (> 1.75 mg/dL) [36]. When the functional requirements are not met by dietary absorption or storage release, serum iron (i.e. the amount of Fe+3 in the blood bound to Tf) decreases while Tf increases. Because of this liaison, three assays that measure the potential of iron supply are generally performed concomitantly, being the serum iron, the Tf concentration (reported as the quantity of iron that can be bound to Tf = total iron binding capacity, TIBC), and the percentage of Tf saturation (serum iron × 100/TIBC) [52]. Serum iron can be measured by either colorimetric assays (most used) or atomic absorption spectrophotometry [53]. The concentration of serum transferrin can be measured by immunologic methods (direct) or throughout the determination of TIBC, whose assay is identical to the serum iron assay, but applies an additional step (saturation of iron-binding sites of the transferrin molecule with excess iron) followed by the removal of the unbound iron. Several analyzers measure also the unsaturated iron binding capacity (UIBC), with TIBC being subsequently calculated by summing UIBC to serum iron [54]. Serum iron, TIBC, and transferrin saturation are indexes of an adequate iron supply, but their utility as screening tools for iron deficiency is limited by several factors, such as the circadian rhythm (e.g., morning peak of serum iron and Tf saturation), diet, and oral contraceptive use [55]. Nevertheless, a Tf saturation < 16% is known to reflect a suboptimal iron supply for the proper erythrocyte development [52]. Normal values of serum iron range between 65 μg/dL to 170 μg/dL in adult males and 50 μg/dL to 170 μg/dL in adult females. TIBC and Tf saturation normal ranges are 250-450 μg/dL and 20-60%, respectively, in both adult males and females [48]. The serum is the best sample matrix, but also heparin-plasma may be used, whilst EDTA- and citrate-plasma are unsuitable due to the chelating properties of these anticoagulants. Cellular ion demands [56], the erythroid proliferation rate [57], and the stainable bone marrow iron [58] are known to be linked to the concentrations of the soluble form of the serine protease-cleaved membrane receptor (sTfR) that circulates in plasma bound to Tf. Several lifestyle factors affect sTfR, such as smoking, alcoholic drinking, sedentary behaviors, and hypernutrition [36]. Latex-enhanced immunoassays (nephelometry and turbidimetry) and the more recent immunofluorometric assays have been implemented to evaluate sTfR. However, the usefulness of commercial kits is limited by the poor comparability between different tools, comprising the calibrators (free vs. transferrin-complexed form, tissue origin), the antibodies (monoclonal vs. polyclonal), and reporting units (mg/L vs. nmol/L) [59]. This lack of commutability together with the relatively high cost of reagents are some of the reasons why sTfR measurements have not been widely adopted in clinical practice. Normal range of sTfR are 0.30-1.75 mg/dL. The serum is the best matrix and it should be separated within 8 h from blood drawings in order to get reliable results [48]. Of note, the sTfR/serum ferritin ratio may be more reliable than each parameter alone for the identification of iron deficiency [60].
- Iron-deficient anemia. The third stage of iron-deficient anemia is characterized by a reduction of both Hb concentrations and RBCs below-optimal levels (i.e. functional iron deficiency = iron supply is inadequate to meet the requirements for erythropoiesis). In the absence of ongoing inflammatory processes, the biochemical features are low ferritin (< 12 μg/L), Tf saturation (< 16%), and Hb (< 115 g/L), but high sTfR (> 1.75 mg/dL) and RBC protoporphyrin (> 80 μg/dL). During the ferrochelatase-dependent insertion of ferrous iron in the proto-porphyrin ring, zinc can alternatively be incorporated to form zinc protoporphyrin, which is normally found in trace amounts [61]. In the early stages of reduced erythropoiesis, erythrocyte zinc protoporphyrin progressively rises, thereby providing to be a useful parameter for detecting uncomplicated functional iron deficiency. Importantly, its measure represents the average iron availability for erythropoiesis during the preceding 3-4 months since they are established during erythrocyte maturation and remain unaltered for the mature RBC lifespan. This value can be measured directly by hematofluorometer (porphyrins fluoresce in the red wavelengths when opportunely excited) or after extraction of the zinc moiety using ethyl acetate and hydrochloric acid. In this latter case, the zinc-free erythrocyte protoporphyrin is measured by conventional fluorometry. Values > 150 μmol/mol heme are highly suggestive of iron deficiency [62]. Although RBCs represent the largest functional compartment, their associated indices are not representative of the individual’s martial status. Hb concentration is usually relevant for assessing the degree of severity of iron deficiency, but its sensitivity is low because of the rather inconsistent variations between healthy and iron-deficient individuals. In addition, the specificity of this test is poor. The packed cell volume (hematocrit, Hct), although widely used in the past, does not provide any additional information to Hb concentration. Altered RBC indices, meaning a reduction of mean corpuscular volume (MCV), a reduction of mean corpuscular hemoglobin (MCH), and an increase of red blood cell distribution width (RDW), are usually a feature of iron-deficient erythropoiesis, but they lack specificity [36, 48]. Conversely, modern analyzers can measure reticulocyte and hypochromic cell parameters, such as the reticulocyte Hb and the proportion of hypochromic erythrocytes, which may be useful for a proper assessment of anemia in chronic conditions characterized by a generalized inflammatory state. For instance, the biochemical feature of functional iron deficiency in chronic heart failure can show normal Hb values [63] and higher cut-off limits for both Tf saturation (< 20%) and ferritin (<300 μg/L) [64]. Heightened values of ferritin may be also found in chronic kidney disease patients, where the concomitant proteinuria, low-iron diet, and inflammation expose them to veiled iron-deficient conditions [65]. The proportion of hypochromic erythrocytes with the reticulocyte Hb count could be used in these cases though, also for predicting the responsiveness to iron therapy [66].”
Revised/new references are:
- Suchdev PS, Namaste SM, Aaron GJ, Raiten DJ, Brown KH, Flores-Ayala R, et al. Overview of the Biomarkers Reflecting Inflammation and Nutritional Determinants of Anemia (BRINDA) Project. Adv Nutr. 2016;7(2):349-356.
- Lynch S, Pfeiffer CM, Georgieff MK, Brittenham G, Fairweather-Tait S, Hurrell RF, et al. Biomarkers of Nutrition for Development (BOND)-Iron Review. J Nutr. 2018;148(suppl_1):1001S-1067S.
- WHO. Iron Deficiency Anaemia - Assessment, Prevention and Control. A guide for programme managers. 2001.
- Munoz M, Villar I, Garcia-Erce JA. An update on iron physiology. World J Gastroenterol. 2009;15(37):4617-4626.
- Level JWHOCfDCaPTCotAoISatP. Assessing the iron status of populations: including literature reviews: report of a Joint World Health Organization/Centers for Disease Control and Prevention Technical Consultation on the Assessment of Iron Status at the Population Level. Geneva, Switzerland, 6-8 April 2004; 2007.
- Harms K, Kaiser T. Beyond soluble transferrin receptor: old challenges and new horizons. Best Pract Res Clin Endocrinol Metab. 2015;29(5):799-810.
- Shih YJ, Baynes RD, Hudson BG, Flowers CH, Skikne BS, Cook JD. Serum transferrin receptor is a truncated form of tissue receptor. J Biol Chem. 1990;265(31):19077-19081.
- Worwood M. Indicators of the iron status of populations: ferritin. Reference Manual for Laboratory Considerations – Iron Status Indicators for Population Assessments. Geneva: WHO; 2007.
- Thurnham DI, McCabe LD, Haldar S, Wieringa FT, Northrop-Clewes CA, McCabe GP. Adjusting plasma ferritin concentrations to remove the effects of subclinical inflammation in the assessment of iron deficiency: a meta-analysis. Am J Clin Nutr. 2010;92(3):546-555.
- Tomkins A. Assessing micronutrient status in the presence of inflammation. J Nutr. 2003;133(5 Suppl 2):1649S-1655S.
- Dignass A, Farrag K, Stein J. Limitations of Serum Ferritin in Diagnosing Iron Deficiency in Inflammatory Conditions. Int J Chronic Dis. 2018;2018:9394060.
- Munoz M, Acheson AG, Auerbach M, Besser M, Habler O, Kehlet H, et al. International consensus statement on the peri-operative management of anaemia and iron deficiency. Anaesthesia. 2017;72(2):233-247.
- Briguglio M, Gianola S, Aguirre MFI, Sirtori P, Perazzo P, Pennestri F, et al. Nutritional support for enhanced recovery programs in orthopedics: Future perspectives for implementing clinical practice. Nutr Clin Metab. 2019;33(3):190-198.
- Lombardi G, Lippi G, Banfi G. Iron requirements and iron status of athletes. In: Maughan RJ, editor. Sports Nutrition. Encyclopaedia of Sports Medicine. XIX. Oxford, UK: John Wiley & Sons; 2014. p. 229-241.
- Gale E, Torrance J, Bothwell T. The quantitative estimation of total iron stores in human bone marrow. J Clin Invest. 1963;42:1076-1082.
- Barron BA, Hoyer JD, Tefferi A. A bone marrow report of absent stainable iron is not diagnostic of iron deficiency. Ann Hematol. 2001;80(3):166-169.
- Zimmermann MB. Methods to assess iron and iodine status. Br J Nutr. 2008;99 Suppl 3:S2-9.
- Beard J. Indicators of the iron status of populations: free erythrocyte protoporphyrin and zinc protoporphyrin; serum and plasma iron, total iron binding capacity and transferrin saturation; and serum transferrin receptor. Reference Manual for Laboratory Considerations – Iron Status Indicators for Population Assessments. Geneve: WHO; 2007.
- Hedayati M, Abubaker-Sharif B, Khattab M, Razavi A, Mohammed I, Nejad A, et al. An optimised spectrophotometric assay for convenient and accurate quantitation of intracellular iron from iron oxide nanoparticles. Int J Hyperthermia. 2018;34(4):373-381.
- Adams PC, Reboussin DM, Leiendecker-Foster C, Moses GC, McLaren GD, McLaren CE, et al. Comparison of the unsaturated iron-binding capacity with transferrin saturation as a screening test to detect C282Y homozygotes for hemochromatosis in 101,168 participants in the hemochromatosis and iron overload screening (HEIRS) study. Clin Chem. 2005;51(6):1048-1052.
- Bullock GC, Delehanty LL, Talbot AL, Gonias SL, Tong WH, Rouault TA, et al. Iron control of erythroid development by a novel aconitase-associated regulatory pathway. Blood. 2010;116(1):97-108.
- Skikne BS, Flowers CH, Cook JD. Serum transferrin receptor: a quantitative measure of tissue iron deficiency. Blood. 1990;75(9):1870-1876.
- Kohgo Y, Niitsu Y, Kondo H, Kato J, Tsushima N, Sasaki K, et al. Serum transferrin receptor as a new index of erythropoiesis. Blood. 1987;70(6):1955-1958.
- Chang J, Bird R, Clague A, Carter A. Clinical utility of serum soluble transferrin receptor levels and comparison with bone marrow iron stores as an index for iron-deficient erythropoiesis in a heterogeneous group of patients. Pathology. 2007;39(3):349-353.
- Speeckaert MM, Speeckaert R, Delanghe JR. Biological and clinical aspects of soluble transferrin receptor. Crit Rev Clin Lab Sci. 2010;47(5-6):213-228.
- Skikne BS, Punnonen K, Caldron PH, Bennett MT, Rehu M, Gasior GH, et al. Improved differential diagnosis of anemia of chronic disease and iron deficiency anemia: a prospective multicenter evaluation of soluble transferrin receptor and the sTfR/log ferritin index. Am J Hematol. 2011;86(11):923-927.
- Labbe RF, Vreman HJ, Stevenson DK. Zinc protoporphyrin: A metabolite with a mission. Clin Chem. 1999;45(12):2060-2072.
- Tillyer ML, Tillyer CR. Zinc protoporphyrin assays in patients with alpha and beta thalassaemia trait. J Clin Pathol. 1994;47(3):205-208.
- Mordi IR, Tee A, Lang CC. Iron Therapy in Heart Failure: Ready for Primetime? Card Fail Rev. 2018;4(1):28-32.
- McDonagh T, Macdougall IC. Iron therapy for the treatment of iron deficiency in chronic heart failure: intravenous or oral? Eur J Heart Fail. 2015;17(3):248-262.
- Kalantar-Zadeh K, Kalantar-Zadeh K, Lee GH. The fascinating but deceptive ferritin: to measure it or not to measure it in chronic kidney disease? Clin J Am Soc Nephrol. 2006;1 Suppl 1:S9-18.
- Ratcliffe LE, Thomas W, Glen J, Padhi S, Pordes BA, Wonderling D, et al. Diagnosis and Management of Iron Deficiency in CKD: A Summary of the NICE Guideline Recommendations and Their Rationale. Am J Kidney Dis. 2016;67(4):548-558..
5) Better describing the "limits" of iron paremteral route treatment in terms of "cost of illness" (hospitalization), adverse effects and metabolic impairment by administrationg an high quantity of elemental iron in short time;
Thanks for pointing out this issue. We integrated the paragraph according to your suggestions.
At line 441: “The prolonged deposit repletion time and impaired absorption render oral supplements vain for patients who require a rapid iron replacement, such as those suffering from heart or kidney disease [63, 105]. Injections of iron-carbohydrate complexes can be the ideal approach, delivering the metal directly into the bloodstream to guarantee the fast replenishment of deposits. The carbohydrate shell helps to isolate the metal from blood components until the complex enters the macrophages of the spleen, the liver, and the bone marrow to be either stored or used. A single dose of intravenous iron may be sufficient to optimize the martial status whereas oral supplements may require daily administrations for weeks [99]. Diverse intravenous iron formulas are available, with differences in unit size, nature of the carbohydrate shell (e.g., dextran, sucrose, gluconate, maltose, sorbitol), surface charge, iron form (Fe+2 or Fe+3) and content [106]. The dose of iron to be administered through parenteral routes can be calculated based on body weight and Hb levels [107], whereas the personalization of oral therapy is often missing, probably due to the perception that the vein infusion is riskier. Indeed, most of the current evidence on safety issues comes from poorly-designed small-scale trials with short follow-ups, possibly concealing long term risks of iron overload or tissue damage, especially for patients undergoing injections with concomitant high ferritin [108]. Despite this widespread mistrust, most of the formulations are safe and supported by a positive benefit-risk ratio when using tailored dosing and monitoring [109, 110], and appears to be more indicated than oral preparations also in conditions of gastrointestinal inflammation or when compliance to oral therapy is dubious. Nevertheless, the diversities in the costs for production, transport, storage, handling (e.g., dilution, contamination risk, in-use stability), and health care assistance render the intravenous preparations not usually considered the first choice of treatment [106, 111].”.
Revised/new references are:
- Winkelmayer WC, Goldstein BA, Mitani AA, Ding VY, Airy M, Mandayam S, et al. Safety of Intravenous Iron in Hemodialysis: Longer-term Comparisons of Iron Sucrose Versus Sodium Ferric Gluconate Complex. Am J Kidney Dis. 2017;69(6):771-779.
- Nikravesh N, Borchard G, Hofmann H, Philipp E, Fluhmann B, Wick P. Factors influencing safety and efficacy of intravenous iron-carbohydrate nanomedicines: From production to clinical practice. Nanomedicine. 2020;26:102178.
- Neogi SB, Devasenapathy N, Singh R, Bhushan H, Shah D, Divakar H, et al. Safety and effectiveness of intravenous iron sucrose versus standard oral iron therapy in pregnant women with moderate-to-severe anaemia in India: a multicentre, open-label, phase 3, randomised, controlled trial. Lancet Glob Health. 2019;7(12):e1706-e1716.
- Del Vecchio L, Longhi S, Locatelli F. Safety concerns about intravenous iron therapy in patients with chronic kidney disease. Clin Kidney J. 2016;9(2):260-267.
- Auerbach M, Macdougall I. The available intravenous iron formulations: History, efficacy, and toxicology. Hemodial Int. 2017;21 Suppl 1:S83-S92.
- New recommendations to manage risk of allergic reactions with intravenous iron-containing medicines. European Medicines Agency's Committee for Medicinal Products for Human Use (CHMP); 2013 13 September 2013. Report No.: EMA/579491/2013.
- Grzywacz A, Lubas A, Fiedor P, Fiedor M, Niemczyk S. Safety and Efficacy of Intravenous Administration of Iron Preparations. Acta Pol Pharm. 2017;74(1):13-24..
6) During the CONCLUSION try to analyze the reason why in 2020 more than one third of the world population still suffering from I.D..(bad patient's compliance to domiciliary treatment with oral iron available: how many patients stopped their treatment with Ferrous Sulphate at home "without speaking with their Medical Doctors? Underdiagnosts of I.D. by Medical Doctors, no drugs or Food supplement available for oral routes with high domiciliary compliance; the ancient Ferrous Sulphate is still the "gold standard" for iron repletion treatment).
We integrated the conclusion by citing the causes most frequently associated with an iron deficiency anemia condition in developed countries. At line 476: “…but iron insufficiency still seems to persist as quite a perplexing and underdiagnosed issue even in developed countries [118]. Even after the diagnosis, either the lack of treatment tailoring or the poor compliance of the patient prevent this condition to be cured [119]. Wellness features like obesity, regular blood donations, or even ethical choices, which lead to consuming strict plant-based diets or contrariwise the most desirable white (low-iron) meat obtained from milk-fed anemic veal calves, are just some of the causes attributable to iron deficiency syndromes. The older the body the more it is exposed to malabsorption syndromes, intestinal bleeding, urinary iron loss, cancer, and polypharmacotherapies [120]. Pragmatic solutions that aim at optimizing the martial status at the population level would be required in the near future, with high-iron foods, oral supplements, or intravenous infusions certainly requiring multimodal and tailored interventions to local conditions and populations of interest”.
Concerning the tip on ferrous sulphate, we integrated some interesting information in the section 5.3. Fortified foods at line 377 “Foods with long shelf lives are therefore fortified with the more stable carbonyl or electrolytic iron powders other than the more soluble ferrous sulphate [78]. These microspheres of pure iron are also known to have high bioavailability [79]” Also at line 422 (5.5. Dietary Supplements) “Of note, comparable bioavailability to ferrous fumarate has been observed for ferrous sulphate [98], the latter still remaining among the most used.”.
New/revised references are:
- Hurrell R, Bothwell T, Cook JD, Dary O, Davidsson L, Fairweather-Tait S, et al. The usefulness of elemental iron for cereal flour fortification: a SUSTAIN Task Force report. Sharing United States Technology to Aid in the Improvement of Nutrition. Nutr Rev. 2002;60(12):391-406.
- Lynch SR, Bothwell T, Powders STFoI. A comparison of physical properties, screening procedures and a human efficacy trial for predicting the bioavailability of commercial elemental iron powders used for food fortification. Int J Vitam Nutr Res. 2007;77(2):107-124.
- Harrington M, Hotz C, Zeder C, Polvo GO, Villalpando S, Zimmermann MB, et al. A comparison of the bioavailability of ferrous fumarate and ferrous sulfate in non-anemic Mexican women and children consuming a sweetened maize and milk drink. Eur J Clin Nutr. 2011;65(1):20-25.118. Mistry R, Hosoya H, Kohut A, Ford P. Iron deficiency in heart failure, an underdiagnosed and undertreated condition during hospitalization. Ann Hematol. 2019;98(10):2293-2297.
- Gebremedhin S, Samuel A, Mamo G, Moges T, Assefa T. Coverage, compliance and factors associated with utilization of iron supplementation during pregnancy in eight rural districts of Ethiopia: a cross-sectional study. BMC Public Health. 2014;14:607.
- Camaschella C. Iron-Deficiency Anemia. N Engl J Med. 2015;373(5):485-486.

Reviewer 3 Report
The text is full of errors and imprecisions - a few are indicated below:
l 73 - whereas ferritin represents about 20% and represents the storage pool, the Tf pool only represents a tiny fraction of this -about 3mg in total, but is the most dynamic compartment of the three which can be readily defined ( functional, storage and transport) , turning over at least 10 times per day.
l 91 - watercress like spinach may contain lots of iron,and ascorbate, but is poorly absorbed on accout of complexation of the frric oiron by phytates and phosphates.
l 119 heme carrier protein is a folate transporter, its role as a low affinity heme transporter has been proposed, but not confirmed
l 28 'pours oxidised iron into the bloodstream' - the consensus is that ferroportin exports ferrous iron which is incorporated into apotransferrin by either membrane-bound heppphaestin or circulating ceruloplasmin.1
l 130 the circulating ferritin acquires its iron within the cell, and plays no role in iron transport
l 147 - ferritin represents an intracellular iron storage pool, not a source of iron
150 the key role of hepcidin in the regulation of systemic iron homeostasis merits a more detailed explanation that the cursory account here.
l 254 biofortification using iron fortified plants has still a veryy long way to go.
Author Response
Response to Reviewers for manuscript nutrients-809963
We would like to thank the Editors and the Reviewer 3 for the assessment of our manuscript. We thank the Reviewer 3 for making suggestions that could improve our article. We welcomed your criticism and corrected the document according to your comments. We believe that now -with the help of the other changes requested by the other Reviewers- our manuscript is highly improved we hope that our revised document addresses all-important issues satisfactorily. Our detailed responses to each suggestion are listed below.
Corrections from Authors to Reviewer 3
The text is full of errors and imprecisions - a few are indicated below:
l 73 - whereas ferritin represents about 20% and represents the storage pool, the Tf pool only represents a tiny fraction of this -about 3mg in total, but is the most dynamic compartment of the three which can be readily defined ( functional, storage and transport) , turning over at least 10 times per day.
We agree with the Reviewer about the tiny fraction of Tf pool. Indeed, we previously reported in Table 1 that “Only 0.1% of total body iron is bound to Tf” and -as we reported at line 71 the grams of total body iron (about 2.3-3.8 g) the calculation is in line with your statement. We have therefore reworded the sentence to make it more consistent with your comment. At line 71 “These mechanisms assure a total body iron of about 2.3 g in women and 3.8 g in men, with almost 60-70% being incorporated in the main circulating protein hemoglobin (Hb), 20% in iron deposits of ferritin, and about 15% in other proteins, such as myoglobin, heme enzymes, transferrin (Tf), and other nonheme compounds [7].”. We added the other suggested information about Tf in another section. Of note, we have decided, following your suggestions and those of another Reviewer, to report more clearly the paragraph “4.1. Overview of Iron Metabolism” in order to make it more understandable. We have therefore included sections to help the reader. Having said that, we have included the information suggested by you at line 135 “Tf binds all iron circulating in plasma and represents the most dynamic compartment, with a turnover rate of about ten times a day that meets the erythropoiesis requirements [21]”.
Revised references are:
- A C Ross, B Caballero, R J Cousins, K L Tucker, Ziegler TR. Modern Nutrition in Health and Disease. 11th ed: Lippincott Williams & Wilkins; 2014.
- Dautry-Varsat A. Receptor-mediated endocytosis: the intracellular journey of transferrin and its receptor. Biochimie. 1986;68(3):375-381.
l 91 - watercress like spinach may contain lots of iron,and ascorbate, but is poorly absorbed on accout of complexation of the frric oiron by phytates and phosphates.
We agree with the comment on iron complexation and its consequent reduction of bioavailability in plants. We have extensively discussed in the manuscript the role of negative effectors (4.2. Absorption Influencers = both negative and positive effectors) both as a single element, such as phytates, and in the context of a whole diet (5.2. Dietary Patterns = differences between Western and plant-based diets). Information on watercress and spinach is inserted at the beginning of the manuscript and in a context in which it focuses on folk medicine. However, we understand the possibility of misunderstanding and we have cross-referenced the section of interest. At line 96 “…greatly favouring iron absorption (see 4.2. Absorption Influencers)…” Of note, Hallberg L & Hulthén L (Am J Clin Nutr. 2000) reported in their paper that the “…addition of 100 mg ascorbic acid to another but similar liquid formula containing 85 mg phytate-P increased iron absorption 3.14 times”, possibly suggesting a superiority of vitamin C over phytates in predicting the bioavailability of iron.
l 119 heme carrier protein is a folate transporter, its role as a low affinity heme transporter has been proposed, but not confirmed
We thank the Reviewer for this clarification. We have corrected the text based on the suggestion. At line 124 “Concerning heme-iron, it is not yet clear how it can be internalized into the enterocyte [17]. The low-affinity heme carrier protein (HCP) has been proposed to have a role, with the metal being subsequently freed from the porphyrin ring by a heme oxygenase [18]”.
Revised references are:
- Gulec S, Anderson GJ, Collins JF. Mechanistic and regulatory aspects of intestinal iron absorption. Am J Physiol Gastrointest Liver Physiol. 2014;307(4):G397-409.
- Le Blanc S, Garrick MD, Arredondo M. Heme carrier protein 1 transports heme and is involved in heme-Fe metabolism. Am J Physiol Cell Physiol. 2012;302(12):C1780-1785.
l 28 'pours oxidised iron into the bloodstream' - the consensus is that ferroportin exports ferrous iron which is incorporated into apotransferrin by either membrane-bound heppphaestin or circulating ceruloplasmin.1
We agree with the Reviewer on what has been said. Our original text also reported the information but differently. However, we support the importance of convey clear information and we have therefore revised the statement to present it more in line with your comment. At line 130 “If iron is required, the basolateral transporter ferroportin (SLC40A1) exports ferrous iron that is subsequently incorporated into apotransferrin by either the membrane-bound hephestin (copper-dependent ferroxidase, so named from “Hephaestus”, the Greek god of metalworking) or the circulating ceruloplasmin (ferroxidase produced by the liver) [20].”.
l 130 the circulating ferritin acquires its iron within the cell, and plays no role in iron transport
We again agree with the Reviewer. The clarification that iron in the blood could also be present in circulating ferritin was not meant to imply that ferritin had a role in transportation. We therefore proceeded to revise the text in order to make it more understandable. At line 136 “The complex Tf-iron interacts with a ubiquitously located receptor and is then internalized through receptor-mediated endocytosis [22]. The subsequent acidification of the vesicle lumen by hydrogen proton pumps allows the offloading of iron-bound Tf and the entry in two different pathways: a recycling pathway, which implies recycling of Tf back to the plasma membrane for iron reloading, and the endosomal degradation pathway, which ends with the release of iron from the endosome thanks to SLC11A2 [23]. Iron can then be sequestered within the deposits of ferritin if it is not required for immediate metabolic functions, such as the synthesis of heme or iron-sulphur clusters [24]. The post-transcriptional regulation of the iron-regulatory protein-1 (IRP1) guarantees the cellular iron homeostasis, which may be inferred from the concentration of circulating ferritin as it is normally secreted by cells in quantities proportional to intracellular deposits. The lesser prevalent hemosiderin is another iron storage complex that less easily releases the metal upon increased requirements [24].”.
l 147 - ferritin represents an intracellular iron storage pool, not a source of iron
We agree with the comment. We clarified the concept. At line 158 “..resident macrophages of the red pulp are in charge of senescent RBCs clearance [25], being capable of metabolizing hemoglobin through proteolysis, heme through heme oxygenase activity, and ferritin through lysosomal degradation. Unless it is not required, the metal exits the macrophages thanks to the SLC40A1, is oxidized by ceruloplasmin and bound to Tf, thus subsequently replenishing most of the Tf iron pool [26]”.
New reference is:
- Nagelkerke SQ, Bruggeman CW, den Haan JMM, Mul EPJ, van den Berg TK, van Bruggen R, et al. Red pulp macrophages in the human spleen are a distinct cell population with a unique expression of Fc-gamma receptors. Blood Adv. 2018;2(8):941-953.
- Ganz T. Macrophages and systemic iron homeostasis. J Innate Immun. 2012;4(5-6):446-453.
150 the key role of hepcidin in the regulation of systemic iron homeostasis merits a more detailed explanation that the cursory account here.
We agree with the comment. At line 163 we dedicated a section to hepcidin. “Other than the abovementioned IRP system, which mainly controls cellular iron uptake and deposits, there is also a general regulatory system for iron homeostasis. Primarily produced by hepatocytes, hepcidin is the master regulator that coordinates dietary absorption, storage, and tissue distribution [27]. Increased hepcidin reduces the number of exposed SLC11 and SLC40, thus blocking the intestinal passage. Consequently, it affects the release of iron from macrophages and hepatocytes, the latter having a great capability for iron deposition in the ferritin form [28]. Reduced iron entry into the bloodstream results in low Tf saturation and lesser iron to be delivered to tissues that expose Tf receptors. Dysregulation of these mechanisms results in iron disorders. Anemia from chronic disease is known to be associated with overexpression of hepcidin, macrophage iron loading, low blood iron, and reduced erythropoiesis [29]. Conversely, negligible hepcidin expression causes higher iron entry into the bloodstream, high Tf saturation, and excess iron accumulation in vital organs (e.g., hemochromatosis) [30].”.
New references are:
- Nemeth E, Ganz T. Regulation of iron metabolism by hepcidin. Annu Rev Nutr. 2006;26:323-342.
- Nunez MT. Regulatory mechanisms of intestinal iron absorption-uncovering of a fast-response mechanism based on DMT1 and ferroportin endocytosis. Biofactors. 2010;36(2):88-97.
- De Domenico I, Ward DM, Kaplan J. Hepcidin regulation: ironing out the details. J Clin Invest. 2007;117(7):1755-1758.
- Anderson GJ. Mechanisms of iron loading and toxicity. Am J Hematol. 2007;82(12 Suppl):1128-1131.
l 254 biofortification using iron fortified plants has still a veryy long way to go..
We agree with the comment. For this and for other information reported in our mini-review, we believe that our manuscript can provide comprehensive and innovative considerations to the topic. We know that we are still in the early stages of biofortification. Indeed, WHO defines it as “Category 3 intervention”, meaning that “available evidence is limited and systematic reviews have not yet been conducted”. We proceeded to insert some new considerations and we corrected the paragraph accordingly. At line 385 “In fact, plants have basic and adaptation mechanisms to incorporate the metal at the root-soil interface (see 2. The Martial Status in Humans) to avoid iron-deficiency symptoms, such as stunted root growth and interveinal chlorosis of young leaves. Biofortification techniques focus on promoting iron incorporation to allow the obtainment of iron-fortified foods [83], but they also aim at obtaining the greatest bioavailability [84]. Despite being a promising agriculture-based approach, there is still limited evidence regarding the clinical efficacy of these biofortified foods to improve nutritional status [85].”.
New references are:
- Sperotto RA, Ricachenevsky FK, Waldow Vde A, Fett JP. Iron biofortification in rice: it's a long way to the top. Plant Sci. 2012;190:24-39.
- Connorton JM, Balk J. Iron Biofortification of Staple Crops: Lessons and Challenges in Plant Genetics. Plant Cell Physiol. 2019;60(7):1447-1456.
- Finkelstein JL, Fothergill A, Hackl LS, Haas JD, Mehta S. Iron biofortification interventions to improve iron status and functional outcomes. Proc Nutr Soc. 2019;78(2):197-207.

Round 2
Reviewer 1 Report
The review paper has been tremendously improved (which is remarkable in such a short time). Most of the comments made in the first review (and all the important ones) have been taken into account and I am happy that these comments have triggered such a thorough and detailed revision.
More than 5 pages and 60 literature references have been added (approx. +50% compared to the previous version), which had allowed the authors to develop appropriately the missing parts. The paper has also been re-structured in several sections, which makes the reading easier. Finally, the English language has been largely improved. This is thus nearly a new paper of much higher quality and interest for the reader.
A few minor comments are below.
Ln 216: prevalence of iron deficiency (not prevalence of anemia)
Ln 328 to 332: use day instead of “die”
Ln 332-333: “The lowest observed adverse effect level is still set at 70.0 mg/die”. Please remove as you have not developed potential adverse effects of iron anywhere else
Ln 404: at the expense of the host ( not “at the expensive of the host”)
Ln 405-407 (about lead pots) should be placed right after mention of iron ingots (ln 398), to gather all that deals with cooking pots.
Ln 412: can you revised the sentence : “Whenever conceivable, it is preferred to resort to dietary supplements to increase the oral iron intake in rural populations”. Dietary supplementation does not only concern rural population and supplements are one possibility among others ( improved diets, fortification) which does not have to be “preferred”
Ln 426-427 : the sentence “In addition, ineffectiveness or other side effects may be avoided [102].” does not bring any information and should be removed
Author Response
Response to Reviewers for manuscript nutrients-809963
We would like to thank the Reviewer 1 for the positive comments. Our responses to your ultimate suggestions are listed below.
Corrections from Authors to Reviewer 1
The review paper has been tremendously improved (which is remarkable in such a short time). Most of the comments made in the first review (and all the important ones) have been taken into account and I am happy that these comments have triggered such a thorough and detailed revision.
More than 5 pages and 60 literature references have been added (approx. +50% compared to the previous version), which had allowed the authors to develop appropriately the missing parts. The paper has also been re-structured in several sections, which makes the reading easier. Finally, the English language has been largely improved. This is thus nearly a new paper of much higher quality and interest for the reader.
A few minor comments are below.
- Ln 216: prevalence of iron deficiency (not prevalence of anemia)
We corrected.
- Ln 328 to 332: use day instead of “die”
We corrected.
- Ln 332-333: “The lowest observed adverse effect level is still set at 70.0 mg/die”. Please remove as you have not developed potential adverse effects of iron anywhere else
We agree with the comment. We removed.
- Ln 404: at the expense of the host ( not “at the expensive of the host”)
Sorry for the mistake. We corrected.
- Ln 405-407 (about lead pots) should be placed right after mention of iron ingots (ln 398), to gather all that deals with cooking pots.
We agree with the proposal. We placed the phrase as suggested.
- Ln 412: can you revised the sentence : “Whenever conceivable, it is preferred to resort to dietary supplements to increase the oral iron intake in rural populations”. Dietary supplementation does not only concern rural population and supplements are one possibility among others ( improved diets, fortification) which does not have to be “preferred”
We agree with the comment. We corrected at line 412 as follows: “People living in poverty may not have access to high-iron foods and pragmatic hands-on approaches are not always implementable in rural areas”.
- Ln 426-427 : the sentence “In addition, ineffectiveness or other side effects may be avoided [102].” does not bring any information and should be removed
We removed at line 427. Yet, we maintained the citation [103] since it is important to refer to the potential misuses and side effects due to the public accessibility of oral supplements with no prescription.

Reviewer 3 Report
I would like to congratulate the authors who have accepted most of my commemts, which essentially related to concerns over the lack of understanding of the multiple facets and the complexity of iron metabolism and homeostasis. I ghave made a few small modification, which I hope that the authors will accept.

Author Response
Response to Reviewers for manuscript nutrients-809963
We would like to thank the Reviewer 3 for the final assessment of our manuscript.
Corrections from Authors to Reviewer 3
I would like to congratulate the authors who have accepted most of my commemts, which essentially related to concerns over the lack of understanding of the multiple facets and the complexity of iron metabolism and homeostasis. I ghave made a few small modification, which I hope that the authors will accept.
We agree with all the suggestions and corrected accordingly to all 6 modifications.
We corrected at line 73, 83, 143, 147.
We removed at line 82, 139.
